# Altered growth conditions more than reforestation counteracted forest biomass carbon emissions 1990–2020

Julia Le Noë [1,2 ✉], Karl-Heinz Erb [1], Sarah Matej [1], Andreas Magerl [1], Manan Bhan[1] & Simone Gingrich [1]

Understanding the carbon (C) balance in global forest is key for climate-change mitigation. However, land use and environmental drivers affecting global forest C fluxes remain poorly quantified. Here we show, following a counterfactual modelling approach based on global Forest Resource Assessments, that in 1990–2020 deforestation is the main driver of forest C emissions, partly counteracted by increased forest growth rates under altered conditions: In the hypothetical absence of changes in forest (i) area, (ii) harvest or (iii) burnt area, global forest biomass would reverse from an actual cumulative net C source of c. 0.74 GtC to a net C sink of 26.9, 4.9 and 0.63 GtC, respectively. In contrast, (iv) without growth rate changes, cumulative emissions would be 7.4 GtC, i.e., 10 times higher. Because this sink function may be discontinued in the future due to climate-change, ending deforestation and lowering wood harvest emerge here as key climate-change mitigation strategies.

[1] Institute of Social Ecology (SEC), Department of Economics and Social Sciences, University of Natural Resources and Life Sciences, Wien, Austria. [2] Present address: Geology Laboratory, École Normale Supérieur, PSL University, Paris, France. ✉email: julia.lenoe@boku.ac.at

Terrestrial ecosystems globally act as net carbon (C) sinks, thus providing a key ecosystem service for global climate-change mitigation[1]. By contrast, global forest biomass has acted as a net C source to the atmosphere over the last three decades, according to the most recent Forest Resource Assessment[2,3]. This global C source arises from the complex interaction of several drivers, operating at different time scales[4], including forest area change, and the balance of gross primary production, respiration in vegetation, and losses through disturbances or extraction[1]. While depletion of C stocks by deforestation and other disturbances is immediate, regrowth is slow and depends on forest age-structure[5] and on the management and climatic conditions affecting growth[6–9]. In the context of climate-change mitigation, it is pivotal to disentangle the roles of these drivers to tap the potentials of different forest-based mitigation strategies[10]. Nevertheless, while several studies isolated the roles of land-use and land-cover change on forest biomass C dynamics[11–14], a consistent comparison of the impacts of different drivers at the global scale has never been performed.

In this work, we fill that gap by combining the most recent and consistent global forest dataset provided by the Forest Resource Assessment (FRA[2])—an authoritative data source[3,15]—with the parsimonious forest C model CRAFT[16] (CaRbon Accumulation in ForesT). This enables us to isolate and quantify the relative impact of various drivers on forest change, including, for the first time, changes in forest growth rates resulting from altered growth conditions. We calculate the temporal dynamics of managed and primary (i.e., unmanaged) forest growth rates in 152 countries in 1990–2020 and couple counterfactual scenario development with a typology approach (see the "Methods" section) in order to answer the following question: Which role do the individual drivers changes in area, harvest, burnt area, and forest growth rate play for the observed C-stock changes in national and global forest biomass dynamics over the last three decades? Answering this question is essential for assessing the efficiency of various forest-based climate-change-mitigation strategies.

## Results and discussion

**Trends in global biomass C stocks**. The CRAFT model reliably reproduces the observed trends in primary and managed forest biomass C stocks (including both above-ground and below-ground biomass) in 1990–2020 with a relative root mean square error (RMSE) of 0.57% between simulated and observed biomass C stocks by the FRA[2] at the global level. These low divergences between stock estimates result, however, in global C emissions c. 2 times lower according to the CRAFT simulations than the estimates derived from the FRA (Supplementary Table 2). Still, the CRAFT simulations corroborated the FRA observations while adding information on annual estimations of forest C stocks, rather than the 5-years interval data provided by the FRA (Fig. 1a), and dynamic annual net C emissions (Fig. 1c, d) from managed and primary forests. The five sensitivity analyses carried out on the most uncertain model inputs and assumptions (see 'Methods' descriptions and Supplementary Figs. 5–10) confirmed the results presented in Fig.1: The largest deviation derived from the sensitivity analysis considering forest gross instead of net area changes results in a relative RMSE of 1.94% with global C emissions c.2.5 times higher than the FRA estimates (Supplementary Table 2). The simulations from the reference model assumptions yield the best RMSE and closest agreement with the C budgets derived from the FRA, indicating that they are the most optimal.

In line with the FRA data, we find here that the main trend is a loss of total biomass C stocks following three phases: increase in annual emissions, stagnation and slight recovery of C stocks,

resulting in net C emissions from forest biomass (Fig. 1c) by 0.74 GtC or 0.03 GtC/yr between 1990 and 2020, contrasted by an opposite trend of increasing biomass density from 70 to 73 tC/ha in total forest (Fig. 1b, d). These figures are within the range of the estimated sink in forest soil and biomass of 0.1 ± 7.3 GtC/yr in 2001–2019 found by Harris et al.[17]. Our estimation is also consistent with that of Tubiello et al.[3] of 0.11 GtC/yr net C emissions from forest ecosystems. A comparison[3] of FRA-derived global forest C emissions with other independent estimates reported in 1990–2015 by National Greenhouse Gas Inventories (NGHGIs)—including the Russian Federation, the USA, China, Indonesia, and India—and by the United Nations Framework Convention on Climate Change for other countries (UNFCCC, 2020[18]) yields a slight difference of c. 18%, although the UNFCCC and NGHGI's account, by definition, only for emissions from managed land[3]. Further independent comparisons at the national and macro-regional levels are compiled in Supplementary Table 1 and reveal that C emissions estimated in the present study are in good agreement with other research.

Here we find that the net C emissions mostly arise from primary forests, which undergo area loss, but also biomass thickening (Fig. 1b, d). By contrast, in spite of area loss, managed forests act as C-sinks following biomass thickening (Fig. 1b, d). Increasing biomass density is therefore key to counteract net C emissions from forest biomass in 1990–2020. While both harvest rate and burnt area increase globally over the period of observation, the increased forest growth rate that we calculate with CRAFT for both primary and managed forests over 1990–2020 emerges here as the only factor explaining increased biomass density at the global level. This is in line with other research pointing to the relevance of biomass thickening for forest C sequestration[19]. In addition, our finding that the forest growth rate increased annually by 0.19%, 0.21%, and 0.21% from 1990 to 2020, respectively, for primary, managed and total forests of the world is consistent with Kolby Smith et al.[20] who find that also net primary production (NPP) increased annually between 0.10 and 0.25% in the period 1982–2011, as well as with other modeling and remote-sensing studies documenting a global greening trend, i.e., vegetation thickening following increased vegetation growth rate[21,22]. Note that estimates of annual growth rate increase in 1990–2020 by the sensitivity analyses provide narrow ranges of 0.17–0.19, 0.21–0.23, and 0.20–0.22%, respectively, for primary, managed, and total forests of the world (Supplementary Table 2).

**Proximate drivers of net C emissions**. We develop six counterfactual scenarios[23–25] in order to investigate how forest biomass density and forest biomass C stocks would evolve in the hypothetical absence of (i) changes in harvest (CF1); (ii) changes in forest growth rates (CF2); (iii) change in burnt area (CF3); (iv) change in forest area (CF4); (v) harvest (CF5); (vi) burnt area (CF6) (see "Methods" section). The comparison of observed and simulated counterfactual trends allows us to isolate and quantify the influence of these four main drivers on global forest C-stock changes at national resolution (CF1 to 4) as well as to quantify the overall effects of total wood extraction and burnt area (CF5 and 6).

At the global level, we find that loss of forest area (CF4) is the main driver of the net C emissions from forest biomass (Fig. 2a). In the absence of changes in area, global forest biomass would act as a cumulative net C sink of c. 26.9 GtC in the study period, creating a difference of 27.6 GtC between the actual and the CF4 C budget. This effect in the absence of area change, however, is a composite of an additional C sink of 30.7 in deforesting countries and an additional C source of 3.8 GtC in reforesting countries.

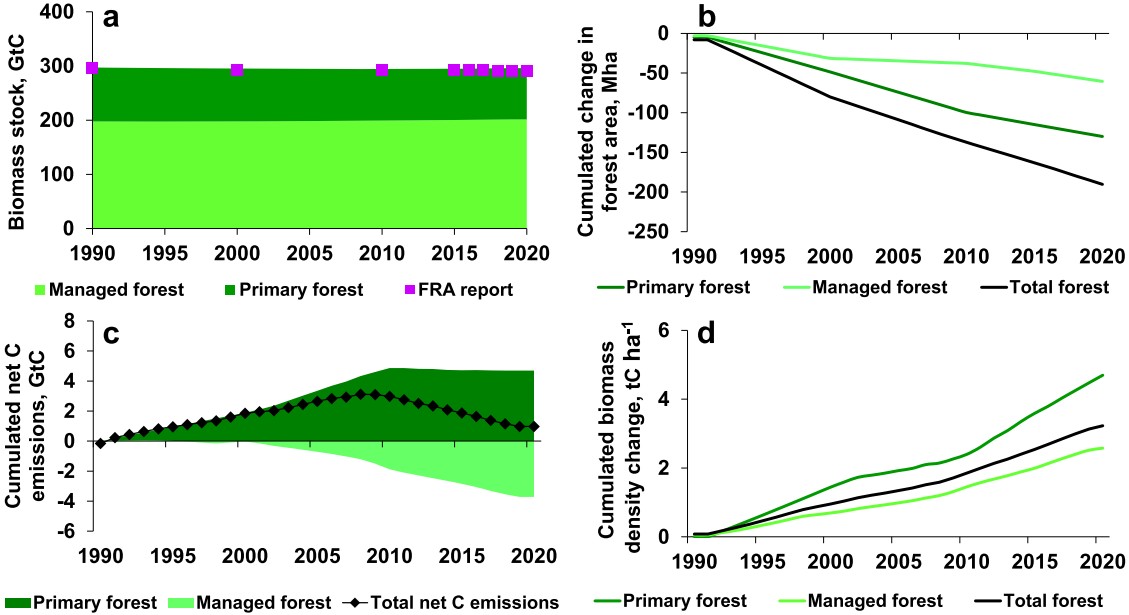

**Fig. 1 Global trends in total, primary, and managed forests. a** Forest biomass C stocks (GtC); **b** cumulated change in forest area (Mha; negative values indicate area loss); **c** cumulated C net emissions (GtC; positive values indicate a C source while negative values indicate a C sink); and **d** cumulated net change in C-stock densities (tC/ha). See Supplementary Fig. 1 for annual fluxes.

Changes in harvest and burnt area from 1990 to 2020 also drove net C emissions from global forest biomass as emissions drop by c. 5.7 and 1.4 GtC in the respective counterfactual scenarios, thus generating net C-sinks of c. 4.9 and 0.63 GtC (Fig. 2a). These figures are in stark contrast with the estimated total sink of c. 49.1 and 5.4 GtC that would emerge in the hypothetical absence of harvest (CF5) and burnt area (CF6; Fig. 2a), respectively. Only changes in forest growth rates counteract the net C emissions from global forest biomass (CF2; Fig. 2a). In the absence of changes in forest growth rates, global forests would act as net C source of c. 7.4 GtC in 1990–2020, i.e., c. 10 times the actually observed source. This net effect in the absence of growth rate change results from an additional C source of 30.4 in countries experiencing growth rate increase and an additional C sink of 23.0 GtC in countries experiencing growth rate decline.

A sensitivity analysis on the potential underestimation of C-dynamics resulting from the use of net area change data at country level (see "Methods" section and Supplementary Fig. 5) reveals that accounting for gross area changes[26] instead of net area change would result in higher global C emissions estimates (4.19 GtC in the sensitivity test versus 0.74 GtC in the reference simulation) but would reveal the same patterns of forest C-dynamic drivers (Supplementary Fig. 5). However, the magnitude of the main drivers would be slightly changed with a lower effect of changes in area (C sink in the hypothetical absence of area changes reaching 20.8 GtC in the sensitivity tests versus 26.9 GtC in the reference assessment) and a higher effect of growth rate changes (C source in the hypothetical absence of growth rate changes reaching 13.1 GtC in the sensitivity tests versus 7.4 GtC in the reference assessment). Generally, the range of results derived from the five sensitivity analyses does not change the relative importance of the individual drivers in any of the scenarios (Fig. 2a, Supplementary Table 3, and Supplementary Fig. 5). However, the sensitivity analyses highlight that the uncertainty is large enough to reverse the cumulated C signal in the absence of changes in harvest (CF1), changes in burnt area (CF3), and the complete absence of burnt areas (CF6). By contrast, the signals of CF2 (no growth rate change), CF4 (no area change), and CF5 (no harvest) are larger than the uncertainty

across sensitivity analyses, signaling that our findings on these drivers are most robust.

The global trends displayed in Fig. 2a, b are the combined results of diverging national forest dynamics (Fig. 2c, h). In particular, shifts in forest area (CF4) contribute to global net C emissions only in the Global South, excluding Vietnam, India, and Chile (Fig. 2f). The impacts of changes in burnt area and harvest are similarly heterogenous, with considerable effects only in some regions (e.g., Vietnam, Mozambique, Fig. 2c, e). In contrast, changes in forest growth rates are more ubiquitous, mainly positive (leading to C-sinks) for most countries, with a few notable exceptions, mainly in arid or boreal regions (e.g., India, Spain, Argentina, Canada; Fig. 2d). Possible reasons explaining the negative effect of change in forest growth rate are forest degradation, increasing drought, cloudiness, or insect outbreaks[15–19]. Over the period 1990–2020, the strongest harvest impacts are observed in countries with large area of managed forest and high harvest pressure, mostly located in temperate and subtropical areas (CF5; Fig. 2g), while fire impacts are strong in only a few countries (CF6; Fig. 2h).

The fact that we use here country-level data comes both with limitations and advantages. The main limitation associated with national data is that it conceals gross C fluxes in forest biomass dynamics and blurs heterogeneity in growth conditions and anthropogenic management within countries. The country-level resolution aggregates the effects of manifold, partly counteracting processes at the local level—including photosynthesis, maintenance respiration, growth respiration, as well as forest area loss and expansion—on the annual dynamic of primary and managed forest biomass. As a consequence, our optimization of the growth function actually reflects apparent national growth rates resulting from the aggregate of these processes. However, this simplification of forest ecosystem functioning is also an advantage. Our approach reproduces forest biomass dynamics very accurately, which is complementary to most process-based models aimed at depicting biological processes and their abiotic controls[27] but providing a wide range of C flux estimations[1] and hardly reproducing observation from inventory data[1,28,29]. By contrast, the strength of the modeling approach implemented here is that it

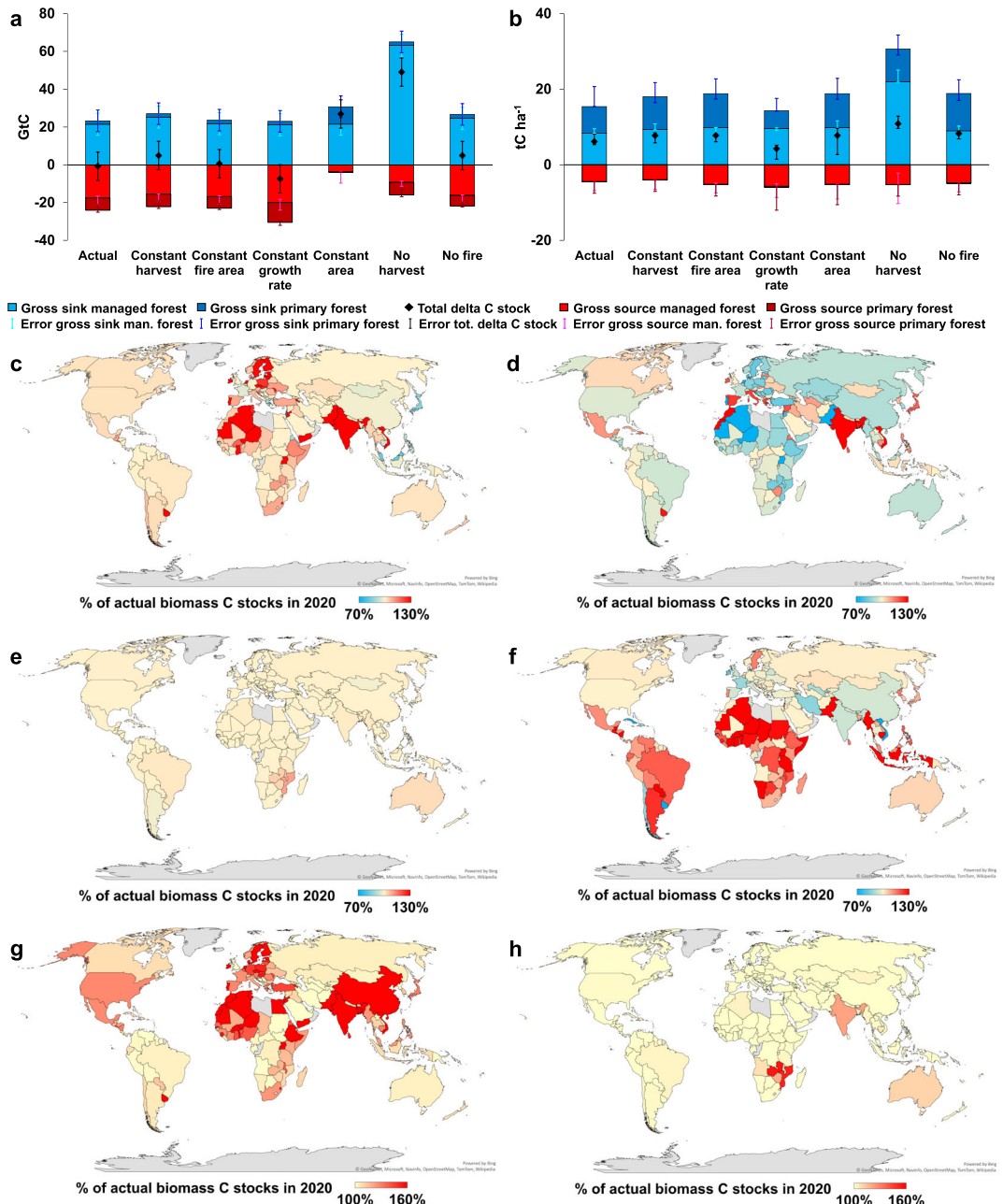

**Fig. 2 Counterfactual scenarios (1990–2020) assessing the cumulative impact of: changes in harvest (CF1); changes in forest growth rate (CF2); changes in burnt area (CF3); changes in forest area (CF4); total harvest (CF5); and total fire (CF6) on C-dynamics.** Panels (**a**) and (**b**) show the global country-level gross and net CF C budgets (GtC) and changes in biomass density (tC/ha), respectively, with negative (red) and positive values (blue) indicating net emissions and sinks, respectively, error bars indicate the range of C budgets estimated across the five sensitivity analyses performed to test the model robustness (see Supplementary Fig. 5 for additional figures showing the net difference between CF and actual C budgets and changes in biomass density, Supplementary Table 3 and Supplementary Fig. 5 for results from sensitivity analyses). Maps show the effects of **c** CF1; **d** CF2; **e** CF3; **f** CF4; **g** CF5; **h** CF6, and are represented as the % of actual biomass C stocks that would be reached in each CF in 2020. Values above 100% (red) indicate that actual change result in net C emissions while values below 100% (blue) indicate that actual change result in a net C sink.

can be run with parsimonious data availability and allows to disentangle the major drivers behind forest C-stock and flux trajectories.

**Typology of forest biomass change**. In order to identify spatial and temporal patterns of drivers in forest biomass trends, we establish a typology of the main drivers over the period 1990–2020 (Fig. 3b). The typology we established is based on the positive versus negative shift in biomass C stocks, and highlights

the most important driver of this shift as assessed through the counterfactual assessment, irrespective of the relative importance of the other drivers shown in Fig. 2. However, as the early separation between increasing and decreasing biomass C stocks in the decision tree (Fig. 3b) may conceal the effect of a major driver counteracting the observed C dynamic, the typology also accounts for possible antagonistic effects by identifying cases in which the main driver of observed C-dynamics is not, in absolute terms, the most important driver (e.g., C stocks increase but the driver with

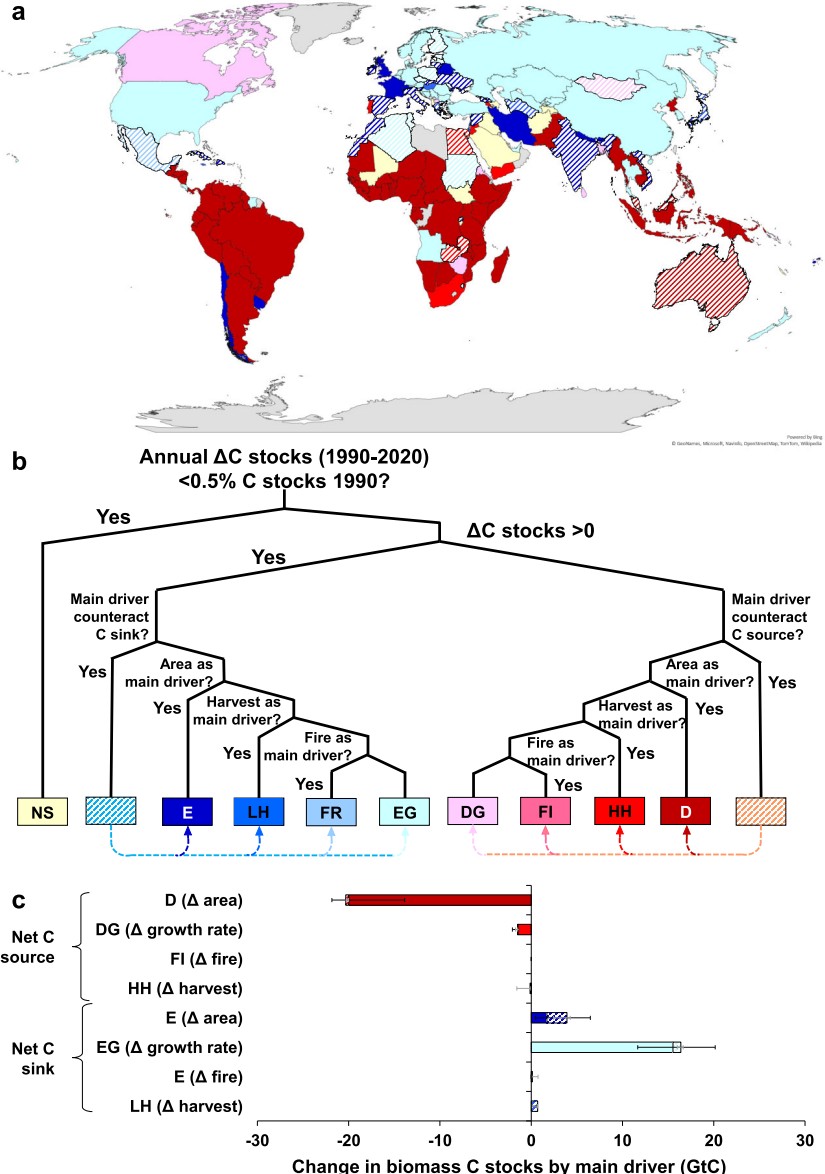

**Fig. 3 Main drivers of the net C emissions from forest biomass. a** Applied at the national level to the 1990–2020 period; **b** established according to a Boolean typology using the results from the counterfactual scenario assessment as criteria; **c** enabling to calculate the sum of net C-sinks and net C sources in each type of forest C-dynamics trajectory identified through the typology, error bars indicating the range of C-sinks and sources by main driver estimated across the five sensitivity analyses, with black and gray bars standing, respectively, for solid and hatched countries (see Supplementary Figs. 6–7 for results from sensitivity analyses). The hatches on the countries (**a**), typology (**b**), and bar chart (**c**) stand for cases in which the driver with the strongest effect actually counteracts the observed carbon budget. The color of the hatches corresponds to the main factor identified by the decision tree algorithm. Abbreviation on the typology: E: C sink driven by forest area Expansion; LH: C sink driven by Lower Harvest; FR: C sink driven by Fire Reduction; EG: C sink driven by Enhanced Growth rate; DG: C source driven by Declining Growth rate; FI: C source driven by Fire Increase; HH: C source driven by Higher Harvest; D: C source driven by Deforestation; NS: non-significant change.

the strongest absolute effect counteracts this positive budget, see also Supplementary Fig. 3). By pinpointing the major drivers of forest change at national levels, such an approach enables to identify major levers for forest conservation.

Deforestation was the dominant driver of net C emissions from forest biomass in most countries of South America and Sub-Saharan Africa, corroborating findings from the literature[11,30,31] (Fig. 3a, c). The net C emissions by countries where deforestation is the most significant driver reach c. 21.3 GtC, with only 0.3 GtC of these emissions being counteracted by another major driver (either increased growth rate or lower harvest pressure). These emissions represent c. 92.7% of the 21.9 GtC net emissions arising

from all countries acting as net C sources (Fig. 3c). Changes in forest growth rates act as the primary drivers in most countries experiencing a net C sink over the period (Fig. 3a, c). The net C-sinks by countries where changes in forest growth rates are the main driver reach c. 16.4 GtC, with 0.9 GtC of these sinks being counteracted by another major driver (increased harvest pressure in all cases except for Sudan where area loss was the major driver counteracting the C sink). These C-sinks mainly driven by increased growth rate represented c. 77.5% of the 21.1 GtC net sink created by all countries acting as net C-sinks (Fig. 3c).

Forest area expansion from 1990 to 2020 is the main driver of forest biomass net C sink in only a few Northern countries but

also some Southern countries, namely Vietnam, India, and Chile, in line with findings reported for these countries[32–34], all together accounting for a net C sink of 3.9 GtC. However, more than half of the C-sinks mainly driven by reforestation are counteracted by another major driver (either declining forest biomass growth rate or increased harvest pressure). Similarly, changes in harvest as well as changes in burnt areas are the main drivers of net C sink or source for a handful of countries in 1990–2020 (Fig. 3a). Finally, declining forest biomass growth rate is the primary driver of net C emissions only in Mongolia and Canada, which is consistent with other studies highlighting slower growth, higher mortality, and insect outbreak events in Canadian forests[35–37].

These highlights derived from the typology remained the same in all sensitivity analyses (Supplementary Figs. 6–7), despite some possible changes in country type identification (Fig. 3a and Supplementary Fig. S6) and amplitude shifts in the attribution of main drivers globally (Fig. 3c and Supplementary Fig. 7). The ranges of values in the attribution of main drivers result from the previously reported differences between the counterfactual and actual C budget estimates across sensitivity analyses (see also Supplementary Tables 2–3) combined with some changes in the type of forest C-dynamics trajectory identified through the typology in countries with large forest biomass stocks: China, India, and Australia (Supplementary Note 1 and Supplementary Fig. 6). However, these shifts do not affect the main conclusions derived from Fig. 3c: in all sensitivity analyses, growth rate changes remain the main driver of global forest biomass C sink with total net C-sinks in countries where increasing growth rate is the main driver (including both solid and hatched countries) ranging from 12.1 to 21.1 GtC, while afforestation always holds the second place of global C sink driver (total net C-sinks in countries where afforestation is the main driver ranging from 2.4 to 7.7 GtC). Similarly, total net C sources by countries where deforestation is the main driver range from −21.9 to −14.0 GtC, thus highlighting that deforestation would by far remain the main driver of forest biomass C emissions across all sensitivity analyses.

**Implications for forest-based solutions**. Our results allow to identify major mechanisms behind observed forest biomass C changes that are immediately relevant for forest-based climate-change-mitigation strategies. We show that deforestation, increasing harvest, and burnt area have driven the net C emissions from forest biomass over the last three decades. Deforestation is the dominant driver, corroborating that protection from deforestation is indispensable[1,11,38]. On the other hand, forest growth rate is identified as the major driver counteracting net C emissions (Fig. 2a, d). In fact, most of the temperate and boreal countries, with the noteworthy exception of Canada, fall under a type in which enhanced forest growth rate is the major driver of a net C sink (Fig. 3b). Besides, even countries dominated by deforestation in the tropics show significant increases in growth rate (Figs. 2d and 4). These results highlight that enhanced growth rate, rather than reforestation, is the main driver counteracting biomass C emissions in 1990–2020.

These increases in forest growth rate may arise from diverse processes, including climatic and land-use drivers. On the one hand, several studies highlight the effects of environmental drivers—such as warming, atmospheric carbon dioxide ($CO_2$), and nitrogen (N) fertilization[1,6,8,11,21,39]—on the terrestrial C sink. On the other hand, changes in forest growth rate can also be driven by shifts in forest management practices, such as tree species selection, forest recovery from past degradation and lesser litter grazing[12,40,41]. Advancing the understanding of the underlying processes of forest growth rate change is key for forging climate-change-mitigation strategies, but it is not straightforward to isolate climatic (e.g., altered $CO_2$

concentration or temperature) from land-use drivers (e.g., non-timber forest uses such as grazing)[42]. Still, a comparison of trajectories in primary and managed forest growth rate change based on our results allows to derive insights into the interplay of these different drivers (Fig. 4 and Supplementary Fig. 3). From the fact that only 11% of primary forest carbon stocks show declining growth rate trends (Fig. 4c) while a relatively larger carbon stock in managed forest (22%) is affected by declining growth rate trends (Fig. 4b), we can infer that in overall terms—and assuming primary and managed forests of a given country to be similarly affected by climatic drivers —land use is likely to exert a degrading effect on growth rate dynamics. Nevertheless, some countries reveal declining growth rate in primary forest but increasing growth rate in managed forest, thus suggesting that forest management may have an improving effect on forest growth rate in those countries (e.g., USA, Fig. 4b, c, see also Supplementary Fig. 4). In overall terms, this result suggests that globally a reduction of forest use may have the potential to enhance growth rate, thus corroborating previous findings by Quesada et al.[14]. However, these interpretations warrant a caveat that primary versus managed forest growth rate changes are derived from the FRA data and a state-of-the-art of the literature on changes in primary forest density (see "Methods" section and Supplementary Note 2), the latter being associated with higher uncertainties although the corresponding sensitivity analysis testing suggests these uncertainties to have little impact on the figures displayed here (see Supplementary Tables 2-3 and Supplementary Figs. 5–10).

Independent of their origin (management or climate driven), the future trajectories of this driver, forest growth rate, is subject to large uncertainties[43–45]. Research suggests that increasing forest growth rate is a transient phenomenon and might be discontinued in the future[46]. For instance, several recent studies have pointed toward the saturating effect of $CO_2$ fertilization, which is suspected to be a key process underlying vegetation greening and ensuing thickening[21], the risk of increasing mortality and slower growth rate following increasing drought[6,47,48], temperature[49], and natural disturbances such as insect outbreaks[50,51]. Even more recently, Duffy et al.[52] showed that, in the near-future, temperature increases from business-as-usual trajectories of climate change shall result in a severe reduction, and possibly a reversal, of the terrestrial C sink, despite the remaining unknowns.

Therefore, we conclude that, while increasing forest growth rate is the dominant driver counteracting the global net C emissions from forest biomass in the past three decades, it is against a precautionary principle to forge climate strategies that rely on a continuous net C sink effect from the same processes in the future. By contrast, our results suggest that reducing wood harvest (Fig. 2g) and halting deforestation (Fig. 2c) are key strategies to address the challenge of climate-change mitigation. In this context, increasing forest harvest volumes—a strategy often promoted in the course of climate-change-mitigation efforts embraced as the "bioeconomy"—appears to have critical unintended side-effects, despite the potential of wood for substituting some emissions-intensive products and processes[53–55]: by not only reducing the carbon sink function in forests, but also accelerating the overall C turnover rates through rejuvenation of forests and transfer to harvested wood products of lifetimes shorter than those of old-growth forests[56–58], such strategies result in a critical loss of C sink capacity. Overall, our results plead for a double strategy to enable future forest-based solutions for climate-change mitigation: in the Global South, ending deforestation is the main priority to reverse the net C source toward a net C sink, while in the Global North, lowering wood harvest has the strongest potential to immediately enhance the C sink in forest biomass.

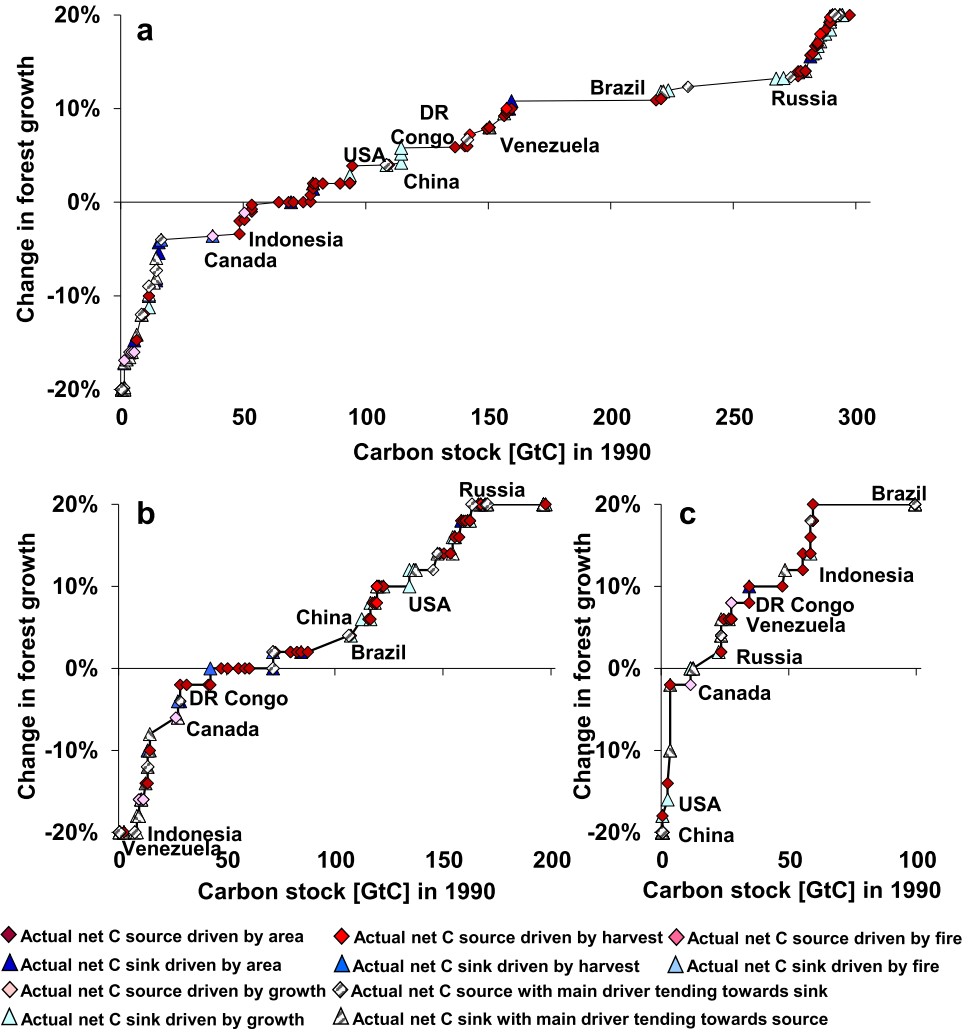

**Fig. 4 Change in forest growth rate and its effects on global carbon stocks.** The diagrams show national forest growth rate changes (y-axis) scaled along the cumulated size of the carbon stock in 1990 (x-axis). The area between the graph and the x-axis indicates the C-stock change due to growth rate for total (**a**), primary (**b**), and managed forests (**c**) (see Supplementary Figs. 8–10 for results from sensitivity analyses).

## Methods

**Principles of the CRAFT model.** CaRbon Accumulation in ForesT (CRAFT) is a parsimonious model of long-term changes in carbon (C) stocks and fluxes in forest ecosystems[16]. CRAFT builds upon the establishment of a place and time-specific relationship between NPP and biomass C stocks and so provides a suitable tool to simulate C-stock dynamics in forest biomass and soils (including both above-ground and belowground biomass). In the present paper, only the biomass module was applied at the national level, distinguishing two key forest categories: primary and managed forest (see definition of the primary versus managed forest below). The input data to simulate C-stock and flux dynamics in biomass are wood harvest (tC), extent of burnt area (Mha), primary, and managed forest area (Mha). The main fixed parameters of the CRAFT model are the mortality rate and the % of losses at cutting, which were taken as 4% and 11%, respectively, based on Liski et al.[59]. The calibration data are the observed data of standing biomass of primary and managed forest at the national level (MtC). From these input and calibration data, we optimized the best growth parameters for each country: $r$ (forest growth rate, $yr^{-1}$) and $K$ (theoretical maximum carrying capacity, $tC\,ha^{-1}$) within a range of possible values (see below). Because the forest growth rate parameter $r$ might be variable over time[9,39,60,61], we assumed that it may linearly increase or decrease from 1990 to 2020. We considered that the value of the $r$ parameter in 2020 could be in a range between 80% and 120% of its estimated value in 2020. All possible combinations for the $r$ and $K$ values in 1990 and changes in $r$ values in 2020 were tested at the national level by steps of 0.01 $yr^{-1}$, 20 $tC\,ha^{-1}$, and 2%, respectively, representing 11,160 possibilities for managed and primary forest of each country. These optimizations were performed with a routine using Macros in Microsoft excel so that simulated data best fit the available observed data within the CRAFT model. At the global level, we calculated forest growth rate values of 0.07, 0.10, and 0.09 $yr^{-1}$ in 1990 and estimated that these values increased annually by 0.19%, 0.21%, and 0.21% until 2020, respectively, for primary, managed, and total forests of the world.

The optimized growth parameters described the relationship between NPP and standing biomass density (B) (Eq. 1) and, together with the other input data, allow to calculate the forest biomass dynamic by recurrence (Eq. 2) such as:

$$NPP_n = rB_n\left(1 - B_n/K\right) \qquad (1)$$

$$B_{n+1} = \left[B_n + NPP_n - (1 + CL)H_{n+1} - FL_{n+1} - mB_n\right] \times MIN(1, A_n/A_{n+1}) \quad (2)$$

With, $n$ the time step (yr); NPP the net primary production ($tC\,ha^{-1}$); $B$ the biomass density ($tC\,ha^{-1}$); $r$ the forest growth rate parameter ($yr^{-1}$), $K$ the theoretical maximum carrying capacity ($tC\,ha^{-1}$); CL the cutting losses (%); $H$ the harvest rate ($tC\,ha^{-1}$); FL the fire losses ($tC\,ha^{-1}$) (see below how the fire losses were estimated); $m$ the mortality rate ($yr^{-1}$); and $A$ the forest area (ha). The initialization was set up by using the calibration data on forest biomass density.

**Range of growth parameter values.** To derive a range of possible values for the forest growth parameters, we relied on-site measurement data on standing biomass and NPP from all ecozones of the world, compiled by Cannell[62], Luyssaert et al.[63], and Anderson-Teixeira et al.[64], contributing 372, 503, and 536 data points, respectively (see Supplementary Table 4). The expansion factors, i.e., the allometric coefficients between tree stems and other tree parts, per ecoregion given in Supplementary Table 5, were used when data reported on biomass missed specific tree organs. Equation (1) was fitted to best reproduce the site measurements in each ecoregion. The $r$ and $K$ parameters of the curve with the smallest residues to the data points were identified (using the MATLAB function fmincon), under the constraints of $r$ being between 0.01 and 1.00, and $K$ being between 0 and 1000. The agreement of fit for the fitted function was calculated. To arrive at a range of $r$ and $K$ values, data were only selectively used for the fitting of the function. For the minimum values of $r$ and $K$, only NPP values below the median were used per 0.2 percentile of biomass values. For the maximum values of $r$ and $K$, only NPP values above the median were used per 0.2 percentile of standing

biomass values. Using this approach, we were able to derive a range of values for the $r$ and $K$ parameters from 0.03 to 0.21 yr$^{-1}$ and from 100 to 720 tC ha$^{-1}$, the lower and upper deciles of the fitted curve. We used these ranges of values for all countries of the world to optimize nationwide growth parameters for both primary and managed forests (see section above).

**Input data forest area.** Data on forest area were extracted from the last Forest Resource Assessment (FRA) (https://fra-data.fao.org/), which distinguishes between 'primary forest', 'mangrove', 'planted forest', and 'naturally regenerated forest'. As mangrove can be both primary and managed forest (FRA[2]), we calculated the primary forest area as the sum of the 'primary forest' area and half of the 'mangrove' area reported by the FRA. As the mangrove area is generally very small in comparison with primary forest area, this assumption does not significantly influence our simulations. The managed forests, i.e., the forests exploited for socio-economic activities, are taken as the total forest area reported by the FRA minus our estimations of primary forest areas.

**Input data harvest.** Wood harvest data were extracted from FAOstat (http://www.fao.org/faostat/en/). We extracted data for eight wood categories: wood fuel, pulpwood, sawlogs, and veneers logs, other industrial roundwood for both coniferous and deciduous wood (provided in m$^3$). To convert wood extraction from m$^3$ to tC, we used coefficients on wood density (t m$^{-3}$), bark fraction (tbark twood$^{-1}$), and C content (tC twood$^{-1}$). These coefficients were specific to product and countries following Haberl et al.[65]. We then summed all wood product harvest at the country level to have an aggregated value of annual wood harvest (tC yr$^{-1}$). As the FAOstat provided wood extraction data only from 1990 to 2019, we assumed the same harvest figures in 2020 as in 2019.

**Input data on burnt area and fire losses.** Data on forest area burnt were extracted from the last FRA (https://fra-data.fao.org/). From this data, we calculated the annual live tree biomass loss by fire such as:

$$FL = \left( C_w \times FL_w \times \alpha_w + C_l \times FL_l \times \alpha_l \right) \times (A_b/A) \times B \quad (3)$$

With FL the annual loss by fire (tC/yr); $C$ the fraction of the respective tree compartment (w = wood, i.e., branches and stem; l = leaves); FL the respective fuel load i.e., the % of live biomass that can actually burn per tree compartment; $\alpha$ the combustion efficiency coefficient, i.e., the % of fuel load consumed by fire per tree compartment in an area affected by fire; $(A_b/A)$, the fraction of area burnt over the total forest area and $B$ the standing biomass (tC). We assumed that the fraction of area burnt was the same in managed and primary forest as in total country-level forest.

Since our study is focused on C in living tree biomass alone, we used stem and foliage as a proxy for living tree biomass, instead of calculating total forest fuels, i.e., understory, litter, duff, deadwood (Supplementary Table 6). We used the assessment by Hoelzemann et al.[66], which provides the percentage of total tree biomass compartment that is available for combustion (i.e., fuel load) per biome (tropical, temperate, boreal Eurasian, American forest). We assessed a range of studies reporting combustion efficiency coefficients for these biomes based on fire models and field observations[67–70] to derive plausible values for combustion efficiency per tree compartment (see Supplementary Table 7). Consequently, we calculated three fire severity levels (low, moderate, high), hence assessing a range of coefficients (fire severity (%) = $\sum_i C_i \times FL_i \times \alpha_i$), which were used to perform sensitivity analyses (see below). We used the moderate fire severity coefficients (ranging from 4 to 11% across world countries) to run the reference model simulations (figures displayed in the main manuscript).

In addition, because the extent of forest area burnt may overlap with deforestation area, we corrected the data provided by the FRA following a 'best guess' approach. We assumed that overlap between forest area burnt and forest area loss was likely to happen in tropical countries[71,72]. By contrast, in boreal and temperate countries fires are rarely purposeful and leading to deforestation[72,73]. Therefore, for tropical forest, we assumed that: if deforestation occurred (i.e., $\Delta A < 0$) and if the extent of fire area was higher than the extent of deforestation (i.e., $A_b - \Delta A > 0$), then the area burnt was corrected by the deforested area, thus assuming that forest area loss was already included in burnt area (i.e., $A_b = A_b - \Delta A$). If deforestation occurred and if the extent of deforestation was higher than the extent of area burnt (i.e., $A_b - \Delta A = <0$), then the area burnt was corrected by zero, thus assuming that the extent of burnt area was already included in the deforested area (i.e., $A_b = 0$). If forest area expansion occurred (i.e., $\Delta A >= 0$), then deforestation cannot overlap with burnt area, thus we applied the data on the extent of forest area burnt as provided by the FRA. For all other countries, we use the data on the extent of area burnt as provided by the FRA without applying correction.

**Calibration data: C stocks in managed and primary forest.** Calibration data on standing biomass C stocks (including both above-ground and belowground biomass) were extracted from the last FRA (https://fra-data.fao.org/). Conversely to the forest area, the FRA did not provide data on biomass C stocks distinguishing between primary and managed forest but only for total forest biomass C stocks at the country level. In the present study, we thus calculated the managed versus primary forest biomass C stocks by using the benchmark values of primary forest

biomass density provided by Erb et al.[12] for the year 2000 at the country level. From this data, we could calculate the biomass C stocks in both primary and managed forest in 2000 such as:

$$B_{prim2000} = BD_{prim2000} \times A_{prim2000} \quad (4)$$

$$B_{man2000} = B_{tot2000} - B_{prim2000} \quad (5)$$

With $B_{prim2000}$, $B_{man2000}$, and $B_{tot2000}$ the C stocks biomass in 2000, respectively, in primary, managed, and total forest. Before and after 2000, forest biomass density of managed and primary forest may follow different trends, as empirical studies revealed that primary forests are not in equilibrium and may undergo changes in biomass density[74] (Supplementary Table 8). In order to consider these changes, we carried out a literature survey to derive temporal trends in boreal, temperate, paleotropical and neotropical primary forest densities (see Supplementary Tables 9–10). Subsequently, we used these macro-regional coefficients to derive biomass C stocks in managed and primary forests at the national level over the 1990–2020 period from the benchmark values of 2000 such as:

$$B_{prim,y} = A_{prim,y} \times BD_{prim,2000} \times \left[ 1 + \delta(2000 - y) \right] \quad (6)$$

$$B_{man,y} = B_{tot,y} - B_{prim,y} \quad (7)$$

With $B_{prim,y}$, $B_{man,y}$, and $B_{tot,y}$ the biomass C stocks (MtC) in primary, managed and total forest in year $y$, with $y$ belonging to [1990–2020]; $A_{prim,y}$ the primary forest area in year $y$ (ha); $BD_{prim,2000}$ the biomass C-stock density (tC ha$^{-1}$) in primary forest in the year 2000 (as provided by Erb et al.[12]); and $\delta$ the annual change in primary forest density (%/yr$^{-1}$) derived from the literature survey (Supplementary Table 10).

**Counterfactual scenario analysis.** In order to isolate and quantify the relative impacts of major proximate drivers on forest biomass C change, we develop six counterfactual scenarios[23–25,75]. In these counterfactual scenarios we investigate how forest biomass density and forest biomass C stocks would evolve in the hypothetical absence of (i) change in harvest (assuming initial average harvest volumes to remain constant; CF1); (ii), change in forest growth rate (calculated values for 1990 remain constant; CF2); (iii) change in burnt area (average values for 2000 to 2003 are assumed to remain constant; note that FRA does not report burnt area before 2000; CF3); (iv) change in forest area (values for 1990 remain constant; CF4); (v) harvest (no-harvest counterfactual; CF5); (vi) fire occurrence (no burnt area counterfactual; CF6). The comparison of observed and simulated trends allows to isolate and quantify the influence of the four main drivers on the global C-stock changes at national resolution (CF1 to 4) as well as to quantify the overall effects of total wood extraction and fire occurrence (CF5 and 6).

**Data uncertainties.** Eighteen countries reported poor data, i.e., incomplete data series or no data regarding forest area, biomass C stocks, extent of fire area or harvest. These countries are Afghanistan, Albania, United Arab Emirates, Azerbaijan, Bosnia and Herzegovina, Western Sahara, Guinea-Bissau, North Macedonia, New Caledonia, Democratic People's Republic of Korea, Rwanda, Saudi Arabia, Sudan, Serbia, South Sudan, Tajikistan, Turkmenistan, Uzbekistan and together represented only 0.64% of the average forest biomass C stocks in 1990–2020. We corrected the incomplete data series for these countries by interpolating the missing data with the one or two closest reported data in the time series. Nevertheless, when data were unavailable over the entire time period, we set the entire timeseries to zero. We warn that the reliability of the results presented for these 18 countries is poor and that further data are required to ascertain the pattern observed in these countries.

By contrast, for the rest of the 134 countries, data quality was very good and sensitivity analyses provided reliable assessments of the uncertainties (see below). According to the last FRA report[2], 86% of the forest area reported was estimated following an IPCC[76] tier 3 approach and 88% of the growing stock trend following tiers 2 or 3 approaches. Similarly, the harvest data reported by the FAO statistics are subject to several statistical tests, data validation, and reconciliation with other data sources[77]. The data reported by the FRA for burnt area arise from the combination of several remote-sensing approaches including the Global Wildfire Information System[78], the Moderate-Resolution Imaging Spectroradiometer (MODIS)[79], and the Global Forest Change product[80]. Nevertheless, even in these 134 countries, the FRA and FAOstat sometimes presented incomplete data series (e.g., data on primary forest areas are provided only from the 2000's in some countries). In such cases, we also corrected the input data by interpolating the missing data with the one or two closest reported data. Data reported on the extent of fire area are the most uncertain. First, because the FRA provided data on the extent of burnt area only from 2000. To fill that gap, we assumed here that the extent of fire from 1990 to 2000 was equal to the average values reported from 2000 to 2003. Second, even in 2000–2020 data gaps on fires exist for some countries. We filled those gaps with the assumption that the extent of fires for the missing years was equal to the average value of the three closest years reported. We explicitly reported all corrections made in the excel file providing input data and model calculations: when any of those assumptions had to be made, data in the input sheets are reported in black, while data extracted from the FAOstat and FRA database are reported in blue (see Source data file 1).

**Sensitivity analyses**. The robustness of the model was assessed through five sensitivity analyses carried out on the most uncertain parameters and assumptions of the model: (i) temporal change in $K$ parameter assumption; (ii) changes in primary forest biomass density assumption; (iii) high fire impact assumption; (iv) low fire impact assumption; (v) effect of gross versus net area changes. As the CRAFT model reproduces the observed trends in forest biomass C stocks, its highest uncertainties lie in the estimation of the growth parameters. Here, two structural hypotheses of the model are likely to affect these calculations: (i) the assumption regarding the optimization of temporal changes in growth parameters, and (ii) the coefficient derived to estimate the temporal trend in primary forest biomass density. To assess the uncertainties associated with those assumptions, we ran 2 sensitivity analyses: (i) We tested an alternative hypothesis in which both the temporal trend of the $r$ and $K$ parameters were optimized against the reference assumption based on the optimization of the temporal change of the $r$ parameter only. (ii) We tested an alternative hypothesis in which biomass density in primary forest was taken as constant (using the benchmark provided in 2000 by Erb et al.[12]) against the reference assumption based on an estimation of changes in primary forest biomass density. Burnt area is another source of uncertainty, because data are only available for the period after 2000 and for some countries missing altogether. Furthermore, it is intricate to separate burnt areas from deforestation areas. We thus consider the fire losses as the most uncertain input data to our model. Therefore, we carried out two additional sensitivity analyses: (iii) we tested the model with fire losses calculated by assuming the lowest fire severity coefficient (see Supplementary Tables 3–4) and by assuming complete overlap between fires and deforestation in all countries (not only in tropical countries, as in the default estimation). (iv) We tested the model with fire losses calculated by assuming the highest fire severity coefficient (see Supplementary Tables 4–5) assuming no overlap between forest area loss and burnt area in any countries. Last, as net area change may result in underestimating C fluxes[81,82] we performed a sensitivity analysis (v) in which annual gross area gain and loss were estimated based on the national conversion factor of gross to net area change derived from Li et al.[26]. To do so, we used the data provided by Li et al.[26] to calculate the average net and average gross national forest area changes in 1992–2015. The ratio of net to gross area change was then derived by dividing the sum of the absolute value of all decreases and increases by the absolute value of the net change. This ratio was then used in the sensitivity analysis in order to quantify the effect of gross area change on forest biomass density (see Eq. 2). The differences in the model outputs and counterfactual scenarios assessments between the five sensitivity analyses and the default model assumptions are presented in Supplementary Tables 2–3 and Supplementary figs. 5–10.

## Data availability

All the data generated in this study are provided in the Source data file 1. Source data are provided with this paper.

## Code availability

The code used to optimize the CRAFT model is provided in the macro included within Source data file 1. Click 'View' in the excel file then click 'Macro' and then 'Edit' to see the code.

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

## Acknowledgements

The authors gratefully acknowledge support by the European Research Council (ERC-2017-StG757995HEFT) and the project CoBALUCE (DFG KA 4815/1-1).

## Author contributions

All authors contributed to the writing of the manuscript and SI. M.B., J.L.N., A.M., and S.M. contributed to literature survey, data compilation, and data processing. K.-H.E., S.G., and J.L.N. designed the paper main ideas and drew the figures. J.L.N. wrote the first and final drafts of the manuscript and Supplementary Information and extended the CRAFT model at the Global level provided in the Source data file 1.

## Competing interests

The authors declare no competing interests.
