## [Peer Review File · Nature Communications]

REVIEWER COMMENTS

Reviewer #1 (Remarks to the Author):

Manuscript: Enhanced growth – more than reforestation – counteracted biomass carbon emissions (1990-2020)

General: This is a well-written manuscript, suitable for publication in *Nature Communications* after some issues have been addressed. My main concern is that the conclusions at the end of the manuscript are not in the abstract (and they should be) and that the title is somewhat misleading. Because young stands 'grow faster' (i.e. enhanced growth) than old stands, it could be interpreted that the study suggests we should harvest primary forests and replace them with younger forests to reduce carbon emissions. The study results do not support this, and it needs to be made clearer that forests accumulate carbon slowly, and harvest/deforestation return that carbon to the atmosphere much more quickly than it was accumulated. First of all, the authors of this study conclude that reduction in harvest and deforestation will be necessary for emissions abatement/continuation of the forest C sink (very important finding!!), but this is somewhat buried at the end of the manuscript. Second, the methods/modeling to determine forest C storage/release are very simple (stock change, no mechanisms, or abiotic controls). I think the estimates/conclusions need to be compared with or put in context with more mechanistic approaches with process-based models. Finally, the fire emissions estimates using the global coefficients are grossly over estimated, especially for coniferous forests 1-4. These need to be much lower given that most of the biomass in a forest is in live trees, and they really don't combust (~5% combustion). There should probably be some references to wood product carbon emissions 5, 6 and substitution benefits 7 in some sections to support the conclusions.

Other:

Line 10: "approached" should be "approach"

Line 82: there is a word missing ("us?")

Line 85/86: This is a very clear finding and it is unsurprising (Deforestation is driving emissions). Shouldn't this be in the abstract more clearly?

Lines 178 to 180 also seem really important. Primary forest is not the reason for 'non-enhanced growth', etc. but this was not mentioned until this point.

Lines 207 onwards: Title and abstract do not reflect this. Many will interpret enhanced growth as 'younger forests grow faster so we need to harvest the old ones and plant young ones'.

Suggested refs:

1. Stenzel, J.E., K.J. Bartowitz, M.D. Hartman, J.A. Lutz, C.A. Kolden, A.M.S. Smith, . . . T.W. Hudiburg, Fixing a snag in carbon emissions estimates from wildfires. *Global Change Biology*, 2019. 0(0). DOI: 10.1111/gcb.14716.
2. Hudiburg, T.W., B.E. Law, W.R. Moomaw, M.E. Harmon, and J.E. Stenzel, Meeting GHG reduction targets requires accounting for all forest sector emissions. *Environmental Research Letters*, 2019. 14(9): p. 095005. DOI: 10.1088/1748-9326/ab28bb.
3. Campbell, J., D. Donato, D. Azuma, and B. Law, Pyrogenic carbon emission from a large wildfire in Oregon, United States. *Journal of Geophysical Research: Biogeosciences*, 2007. 112(G4): p. G04014. DOI: 10.1029/2007JG000451.
4. Ghimire, B., C.A. Williams, G.J. Collatz, and M. Vanderhoof, Fire-induced carbon emissions and regrowth uptake in western U.S. forests: Documenting variation across forest types, fire severity, and climate regions. *Journal of Geophysical Research: Biogeosciences*, 2012. 117(G3). DOI: doi:10.1029/2011JG001935.
5. Hudiburg, T.W., B.E. Law, W.R. Moomaw, M.E. Harmon, and J.E. Stenzel, Meeting regional GHG reduction targets requires accounting for all forest sector emissions. *Environmental Research Letters*, 2019. 14(9): p. 095005.
6. Dymond, C.C., Forest carbon in North America: annual storage and emissions from British Columbia's harvest, 1965–2065. *Carbon Balance and Management*, 2012. 7: p. 8-8. DOI: 10.1186/1750-0680-7-8.
7. Harmon, M.E., Have product substitution carbon benefits been overestimated? A sensitivity analysis of key assumptions. *Environmental Research Letters*, 2019. 14(6): p. 065008.

Reviewer #2 (Remarks to the Author):

Enhanced growth – more than reforestation – counteracted biomass carbon emissions (1990-2020), by Julia Le Noë et al.

The manuscript by Le Noë et al. assesses drivers of changes in global forest biomass for 1990-2020 using an approach that simulates biomass changes at country level. For this, national statistics from FAO's Forest Resource Assessment are used in a simple model setup representing the main processes and drivers of forest biomass change.

The study addresses an important question with relevance to the broad scientific audience, and the topic fits well with the strong focus on forest biomass change as a means of climate change mitigation. The use of FAO's national data in the study is a welcome supplement to the literature, which is at these scales otherwise dominated by data sets based on gridded inventory data, remote sensing-derived estimates and (gridded) vegetation modelling.

I have some questions about the methods, and in particular about the implications of the use of these FAO statistical data at country level. Apart from that, I am not very convinced about the typology used, as it may promote drivers that are not the largest single effects in case two or more other drivers oppose this change (details below).

Having said that, I think the study is a very welcome contribution to the ongoing discussions about changes in the forest carbon sink and I warmly recommend publication of this study once my concerns below are addressed.

I provide detailed comments below to aid revisions of the manuscript.

I think that the fact that drivers (in the counterfactual simulations) are assessed as aggregated to national level should be better reflected in the discussion. E.g., when showing totals disaggregated into increase and decrease (e.g., Fig. 2ab), these increase and decrease are based on national numbers, whereas there also may be a composite of increase and decrease within each country.

Although I see the importance of some sort of typology to summarize main drivers, I am in doubt whether the current scheme (supplementary material S3) is accurate. By separating early on in the scheme between increase and decrease, one can easily end up in a situation where individual factors with the largest change do not show up, e.g. with a large increase or decrease in area counterbalanced by two other factors (e.g. changes in growth or in harvest) – if the sum of the latter exceeds that of the former, it is one of the latter factors that will be highlighted.

Also, I would like to suggest moving the explanation of the typology (or a short description of it) to the main text. It is key to understand Fig. 3, but at the moment "hidden" in the supplementary material.

"On the one hand, several studies highlight the feedback of environmental variables...": The effects mentioned here are drivers, not feedbacks.

Fig. 4 caption: This is not a Marimekko diagram – it lacks the spatial ("Marimekko") pattern of it. But it is a nice attempt to visualize the data. Does it contain all 152 countries? The number of points seems limited, but this may also be because data are rounded to whole percent numbers. A slightly larger version of the figure would help with the annotation of countries (which I think is very informative here). And you could consider to draw the rectangles between the line and the x-axis to highlight area changed. You could also decide to use colours again in your annotation to highlight the importance of the different drivers.

The method, and with that the outcome of the study, depend greatly on the use of the FRA data, and on the

assumptions made for that. I would welcome a more thorough discussion of the results and the uncertainties that are introduced by these data. Some questions and thoughts that could be relevant for that:

- The aggregation of data to country level makes that in some cases greatly different areas/types of forest are lumped together. Are the assumed uniform changes within the country (one growth rate, one total biomass, one rate of afforestation/deforestation, etc.) accurate?
- Eq. 2: In the last part, $\text{MIN}(1, A(n)/A(n+1))$ seems to indicate that biomass density is assumed to decrease with increasing area (to account for the establishment of new forest with low initial biomass, I presume). Does the growth function (Eq. 1) capture the biomass increase accurately when doing so? This assumes basically that the existing biomass is "diluted" over a larger area, but does this give a good average of the unequal growth between the "old", established area and the added area with more rapid growth of new forest?
- How do the incomplete data on forest fires affect the estimated role of fire? What is the risk that this is now erroneously accounted for in e.g. altered linear growth trends?

Input data forest area: From the given description, "production forest" contains both planted and naturally regenerated forest (secondary forest). How large are these two contributions to the total of "production forest"? If the secondary forest is a significant contributor, it may need a more inclusive title, as secondary forest is not necessarily a production forest.

REVIEWER COMMENTS

Dear editor, thank you very much for giving us the opportunity to revise our manuscript. The reviewer comments were very helpful and constructive and helped us to improve the clarity of our descriptions and the messages conveyed. In particular:

- We performed an advanced sensitivity analysis, as all reviewers remarked that data quality needs to be reflected, including an exploration of the effect of gross versus net area changes on our findings.
- We further put our estimates and findings in context with the literature, including a supplementary table comparing our biomass C budget estimates with other independent research.
- We also reflected on the remark made by reviewer 1 regarding the extent of fire impact. We figured out that we previously erroneously considered coefficients reported for litter and dead wood while our focus is here on live biomass only. Consequently, we have re-calculated the fire losses component after carefully revising the coefficients of fire severity. This modification implied to entirely re-run the model and re-do all figures. This modification slightly changed our overall result and down-graded the role of fire. However, this did not affect our conclusions, in particular the main finding regarding the pivotal role of changes in growth conditions and changes in forest areas.
- We reconsidered and improved the typology of forest biomass carbon budget main driver following reviewer 2 remark.
- We revised the text in the main manuscript and SI accordingly.

Importantly, the findings of our study, e.g., growth-condition-changes being more important than area increase, remain unaltered by this revision of the empirical part. We are very confident that we now present an even stronger analysis, as the revision allowed us to directly address the impact of data and concept uncertainties. Again, we are very grateful for this opportunity to revise and improve our manuscript.

Reviewer #1 (Remarks to the Author):

Manuscript: Enhanced growth – more than reforestation – counteracted biomass carbon emissions (1990-2020)

Q.1.1. General: This is as well-written manuscript, suitable for publication in Nature Communications after some issues have been addressed. My main concern is that the conclusions at the end of the manuscript are not in the abstract (and they should be) and that the title is somewhat misleading. Because young stands 'grow faster' (i.e. enhanced growth) than old stands, it could be interpreted that the study suggests we should harvest primary forests and replace them with younger forests to reduce carbon emissions. The study results do not support this, and it needs to be made clearer that forests accumulate carbon slowly, and harvest/deforestation return that carbon to the atmosphere much more quickly than it was accumulated.

A.1.1. We thank the reviewer for the constructive remarks and appreciation of our manuscript. We have done our best to reply to all comments addressed by the reviewer (see our reply below). In addition, we would like to mention that reviewer 2's remark on the typology decision tree made us reconsider and improve the typology (see Fig. 3 and the 'Typology of forest biomass change' section in the main manuscript).

Thanks for drawing our attention on the potential confusion in the title. We modified the title as follows 'Altered growth conditions more than reforestation counteracted forest biomass carbon emissions 1990-2020'.

We also modified the abstract to make clear that the faster growth we observed results not from younger stands but from environmental and management changes and to make our final conclusion sharper. The abstract now reads as follows:

Lines 7-21 (revised manuscript track change)

“Understanding the carbon (C) balance in global forest is key for climate-change mitigation. However, land use and environmental drivers affecting global forest C fluxes remain poorly quantified. Following a counterfactual modelling approach based on global Forest Resource Assessments, we show that in 1990-2020 deforestation was the main driver of forest C emissions, partly counteracted by increased forest growth rates under altered conditions: In the hypothetical absence of changes in forest (i) area, (ii) harvest or (iii) burnt area, global forest biomass would have reversed from an actual cumulative net C source of c. 0.74 GtC to a net C sink of 26.9, 4.9 and 0.63 GtC, respectively. In contrast, (iv) without growth rate changes, cumulative emissions would be 7.4 GtC, i.e. 10 times higher. Because this sink function may be discontinued in the future due to climate-change, ending deforestation and lowering wood harvest emerge as key climate-change mitigation strategies.”

Q.1.2. First of all, the authors of this study conclude that reduction in harvest and deforestation will be necessary for emissions abatement/continuation of the forest C sink (very important finding!!), but this is somewhat buried at the end of the manuscript.

A.1.2. In order to make the final message more explicit, clearer, and more impactful in the end of the manuscript we removed the following sentence: “The tradeoffs between the wood provisioning and regulating ecosystem services should, however, be considered in all their complexity, because declining wood harvest might occur together with wood product substitution by other materials generating additional emissions”.

We also added a clearer statement to the abstract that strengthened the final message and added references to support it in the manuscript. The final paragraph now reads as follows:

Lines 321-340 (revised manuscript track change)

“Therefore, we conclude that, while increasing forest growth rate is the dominant driver counteracting the global net C emissions from forest biomass in the past three decades, it is against a precautionary principle to forge climate strategies that rely on a continuous net C sink effect from the same processes in the future. By contrast, our results suggest that reducing wood harvest (Fig. 2g) and halting deforestation (Fig.2c) are key strategies to address the challenge of climate-change mitigation. In this context, increasing forest harvest volumes – a strategy often promoted in the course of climate change-mitigation efforts embraced as the “bioeconomy” – , appears to have critical unintended side-effects, despite the potential of wood for substituting some emissions-intensive products and processes⁵³⁻⁵⁵: By not only reducing the carbon sink function in forests, but also accelerating the overall C turnover rates through rejuvenation of forests and transfer to harvested wood products of lifetimes shorter than those of old-growth forests⁵⁶⁻⁵⁸, such strategies result in a critical loss of C sink capacity. Overall, our results plead for a double strategy to enable future forest-based solutions for climate-change mitigation: in the Global South, ending deforestation is the main priority to reverse the net C source towards a net C sink, while in the Global North, lowering wood harvest has the strongest potential to immediately enhance the C sink in forest biomass.”

References

53. Law, B. E. *et al.* Land use strategies to mitigate climate change in carbon dense temperate forests. *Proc. Natl. Acad. Sci.* **115**, 3663–3668 (2018).
54. Braun, M. *et al.* A holistic assessment of greenhouse gas dynamics from forests to the effects of wood products use in Austria. *Carbon Manag.* **7**, 271–283 (2016).
55. Harmon, M. E. Have product substitution carbon benefits been overestimated? A sensitivity analysis of key assumptions. *Environ. Res. Lett.* **14**, 065008 (2019).
56. Erb, K.-H. *et al.* Biomass turnover time in terrestrial ecosystems halved by land use. *Nat. Geosci.* **9**, 674–678 (2016).
57. Hudiburg, T. W., Law, B. E., Moomaw, W. R., Harmon, M. E. & Stenzel, J. E. Meeting GHG reduction targets requires accounting for all forest sector emissions. *Environ. Res. Lett.* **14**, 095005 (2019).
58. Dymond, C. C. Forest carbon in North America: annual storage and emissions from British Columbia’s harvest, 1965–2065. *Carbon Balance Manag.* **7**, 8 (2012).

Q.1.3. Second, the methods/modeling to determine forest C storage/release are very simple (stock change, no mechanisms, or abiotic controls). I think the estimates/conclusions need to be compared with or put in context with more mechanistic approaches with process-based models.

A.1.3. As the reviewer correctly points out, the CRAFT model used in the present study is a simple empirical and statistical approach aiming at reproducing inventory report data on forest biomass C stocks, which it achieves very well with an average relative root mean square error between simulated and observed biomass C stocks of 0.57% at the global scale. Therefore, the uncertainties regarding the estimates of forest biomass C stocks and fluxes mostly reflect the uncertainties of the Forest Resource Assessment (FRA) database developed by the FAO which is used in the present study for both calibration and validation. To answer this important remark regarding the assessment of model, data and results robustness and reliability we made 2 types of revisions in the manuscript and SI:

- 1- We have thoroughly revised the passages on data and concept uncertainty, we now discuss model uncertainty in the main manuscript and put systematically our results in context with results obtained by other approaches. In particular, (i) we carried out a literature survey in order to provide independent comparisons between our modelled estimates of forest biomass carbon budgets and estimates by other independent studies (summarized in Table S1). The comparisons are very satisfying as our estimates are in good agreement with all other independent researches. (ii) We also compared our estimates with that of the FRA (Table S2) and show that the model simulation allows to accurately reproduce the forest biomass C stocks reported by the FRA, even though the very small divergences between biomass C stock result in more significant discrepancies in the global C budget (net emissions of 0.74 GtC according to the CRAFT simulations versus 1.69 GtC derived from the FRA). (iii) Last, we confronted our main findings – changes in growth conditions are key driver counteracting forest C emissions – with other modelling studies. To that end we further dug into the literature and inserted several sentences and paragraphs emphasizing that other modelling and remote-sensing studies also found an important role of changes in growth conditions. However, no study is available that systematically scrutinized the role of individual drivers of C-stock changes, such as area and harvest changes, in concomitance with changes in growth conditions. To allow for the latter, isolate changes in growth condition, e.g. from rejuvenation effects, is a key aspect of the CRAFT model used in the present study.
- 2- In order to assess the robustness of our modelling approach and results, we carried out 5 sensitivity analyses on the most uncertain parameters/assumptions: (i) constant primary forest biomass density assumption (ii) temporal change in K parameter assumption; (iii) high fire impact assumption; (iv) low fire impact assumption; (v) effect of gross versus net area changes. These sensitivity analyses allow us to provide error bars where relevant in the manuscript figures (Fig. 2a & b, Fig. 3c). The main figures arising from these five sensitivity analyses are also displayed in the Supplementary Information (see Tables S2-3 and Fig. S5-10). Overall the comparison between the results presented in the main manuscript *versus* those displayed in the SI as part of the sensitivity analysis confirmed the robustness of the model and the reliability of our main conclusions: there were only differences in the magnitudes of the estimates by the various sensitivity tests but the

results from the sensitivity analyses did not affect our main conclusion that changes in growth conditions more than reforestation counteracted the biomass C emissions mainly caused by deforestation in 1990-2020.

We revised the manuscript accordingly. The additional paragraphs inserted in the main manuscript read as follows:

Lines 64-70 (revised manuscript track change, see also lines 693-731 quoted below)
Commenting the sensitivity analysis assessment:

“The five sensitivity analyses carried out on the most uncertain model inputs and assumptions (see method descriptions and Fig.S5-10) confirmed the results presented in Fig.1: The largest deviation derived from the sensitivity analysis considering forest gross instead of net area changes results in a relative RSME of 1.94% with global C emissions c.2.5 times higher than the FRA estimates (Table S2). The simulations from the reference model assumptions yield the best RMSE and closest agreement with the C budgets derived from the FRA, indicating that they are the most optimal.”

Lines 76-86 (revised manuscript track change) *Putting our estimates in context with other researches:*

“These figures are within the range of the estimated sink in forest soil and biomass of 0.1 ± 7.3 GtC/yr in 2001-2019 found by Harris et al¹⁷. Our estimation is also consistent with that of Tubiello et al³ of 0.11 GtC/yr net C emissions from forest ecosystems. A comparison³ of FRA-derived global forest C emissions with other independent estimates reported in 1990-2015 by National Greenhouse Gas Inventories (NGHGs) – including the Russian Federation, the USA, China, Indonesia and India – and by the United Nations Framework Convention on Climate Change for other countries (UNFCCC, 2020¹⁸) yields a slight difference of c.18%, although the UNFCCC and NGHGI’s account, by definition, only for emissions from managed land³. Further independent comparisons at the national and macro-regional levels are compiled in Table S1 and reveal that C emissions estimated in the present study are in good agreements with other researches.”

References

3. Tubiello, F. N. *et al.* Carbon emissions and removals from forests: new estimates, 1990–2020. *Earth Syst. Sci. Data* **13**, 1681–1691 (2021).
17. Harris, N. L. *et al.* Global maps of twenty-first century forest carbon fluxes. *Nat. Clim. Change* (2021) doi:10.1038/s41558-020-00976-6.
18. UNFCCC: United Nations Framework Convention on Climate Change: *Time Series – Annex I, UNFCCC*. https://di.unfccc.int/time_series (Bonn).

Lines 95-104 (revised manuscript track change) *Putting our main finding in context with other researches:*

“This is in line with other research pointing to the relevance of biomass thickening for forest C sequestration¹⁹. In addition, our finding that the forest growth rate increased annually by 0.19, 0.21 and 0.21 % from 1990 to 2020 respectively for primary, managed and total forests of the world is consistent with Kolby Smith et al.²⁰ who find that also Net Primary Production (NPP) increased annually between 0.10-0.25% in the period 1982-2011, as well as with other modelling

and remote-sensing studies documenting a global greening trend, i.e. vegetation thickening following increased vegetation growth rate^{21,22}. Note that estimates of annual growth rate increase in 1990-2020 by the sensitivity analyses provide narrow ranges of 0.17-0.19, 0.21-0.23 and 0.20-0.22% respectively for primary, managed and total forests of the world (Table S2).”

References

19. Kauppi, P. E. *et al.* Carbon benefits from Forest Transitions promoting biomass expansions and thickening. *Glob. Change Biol.* **26**, 5365–5370 (2020).
20. Kolby Smith, W. *et al.* Large divergence of satellite and Earth system model estimates of global terrestrial CO₂ fertilization. *Nat. Clim. Change* **6**, 306–310 (2016).
21. Zhu, Z. *et al.* Greening of the Earth and its drivers. *Nat. Clim. Change* **6**, 791–795 (2016).
22. Zhao, L., Dai, A. & Dong, B. Changes in global vegetation activity and its driving factors during 1982–2013. *Agric. For. Meteorol.* **249**, 198–209 (2018).

Lines 138-153 (revised manuscript track change) Commenting the sensitivity analysis assessment:

“A sensitivity analysis on the potential underestimation of C dynamics resulting from the use of net area change data at country level (see methods section and Fig. S5) reveals that accounting for gross area changes²⁶ instead of net area change would result in higher global C emissions estimates (4.19 GtC in the sensitivity test versus 0.74 GtC in the reference simulation) but would reveal the same patterns of forest C-dynamic drivers (Fig. S5). However, the magnitude of the main drivers would be slightly changed with a lower effect of changes in area (C sink in the hypothetical absence of area changes reaching 20.8 GtC in the sensitivity tests versus 26.9 GtC in the reference assessment) and a higher effect of growth rate changes (C source in the hypothetical absence of growth rate changes reaching 13.1 GtC in the sensitivity tests versus 7.4 GtC in the reference assessment). Generally, the range of results derived from the five sensitivity analyses does not change the relative importance of the individual drivers in any of the scenarios (Table S3, Fig. 2a and S5). However, the sensitivity analyses highlight that the uncertainty is large enough to reverse the cumulated C signal in the absence of changes in harvest (CF 1), changes in burnt area (CF 3), and the complete absence of burnt areas (CF 6). By contrast, the signals of CF 2 (no growth rate change), CF 4 (no area change) and CF 5 (no harvest) are larger than the uncertainty across sensitivity analyses, signaling that our findings on these drivers are most robust.”

Reference

26. Li, W. *et al.* Gross and net land cover changes in the main plant functional types derived from the annual ESA CCI land cover maps (1992–2015). *Earth Syst. Sci. Data* **10**, 219–234 (2018).

Lines 167-180 (revised manuscript track change) Reflecting on the scale issue implication:

“The fact that we use here country-level data comes both with limitations and advantages. The main limitation associated with national data is that it conceals gross C fluxes in forest biomass dynamics and blurs heterogeneity in growth conditions and anthropogenic management within countries. The country-level resolution aggregates the effects of manifold, partly counteracting processes at the local level – including photosynthesis, maintenance respiration, growth respiration, as well as forest area loss and expansion – on the annual dynamic of primary and managed forest biomass. As a consequence, our optimization of the growth function actually

reflects apparent national growth rates resulting from the aggregate of these processes. However, this simplification of forest ecosystem functioning is also an advantage. Our approach reproduces forest biomass dynamics very accurately, which is complementary to most process-based models aimed at depicting biological processes and their abiotic controls²⁷ but providing a wide range of C flux estimations¹ and hardly reproducing observation from inventory data^{1,28,29}. By contrast, the strength of the modelling approach implemented here is that it can be run with parsimonious data availability and allows to disentangle the major drivers behind forest C stock and flux trajectories.”

References

1. Friedlingstein, P. *et al.* Global Carbon Budget 2020. *Earth Syst. Sci. Data* **12**, 3269–3340 (2020).
27. Pongratz, J. *et al.* Models meet data: Challenges and opportunities in implementing land management in Earth system models. *Glob. Change Biol.* **24**, 1470–1487 (2018).
28. Bellassen, V. *et al.* Reconstruction and attribution of the carbon sink of European forests between 1950 and 2000. *Glob. Change Biol.* **17**, 3274–3292 (2011).
29. Ciais, P. *et al.* Carbon accumulation in European forests. *Nat. Geosci.* **1**, 425–429 (2008).

Lines 231-246 (revised manuscript track change) Commenting the sensitivity analysis assessment:

“These highlights derived from the typology remained the same in all sensitivity analyses (Fig. S6-7), despite some possible changes in country type identification (Fig. 3a and S6) and amplitude shifts in the attribution of main drivers globally (Fig. 3c and S7). The ranges of values in the attribution of main drivers result from the previously reported differences between the counterfactual and actual C budget estimates across sensitivity analyses (see also Tables S2-3) combined with some changes in the type of forest C-dynamics trajectory identified through the typology in countries with large forest biomass stocks: China, India and Australia (Fig. S6 and explanations below Fig. S6). However, these shifts do not affect the main conclusions derived from Fig. 3c: In all sensitivity analyses, growth rate changes remain the main driver of global forest biomass C sink with total net C sinks in countries where increasing growth rate is the main driver (including both solid and hatched countries) ranging from 12.1 to 21.1 GtC, while afforestation always holds the second place of global C sink driver (total net C sinks in countries where afforestation is the main driver ranging from 2.4 to 7.7 GtC). Similarly, total net C sources by countries where deforestation is the main driver range from -21.9 to -14.0 GtC, thus highlighting that deforestation would by far remain the main driver of forest biomass C emissions across all sensitivity analyses.”

Lines 572-578 (revised manuscript track change) Further discussing data uncertainty:

“According to the last FRA report², 86% of the forest area reported was estimated following an IPCC⁷⁶ tier 3 approach and 88% of the growing stock trend following tiers 2 or 3 approaches. Similarly, the harvest data reported by the FAO statistics are subject to several statistical tests, data validation and reconciliation with other data sources⁷⁷. The data reported by the FRA for burnt area arise from the combination of several remote-sensing approaches including the Global Wildfire Information System⁷⁸, the Moderate-Resolution Imaging Spectroradiometer (MODIS)⁷⁹ and the Global Forest Change product⁸⁰.”

References

2. *Global Forest Resources Assessment 2020*. (FAO, 2020). doi:10.4060/ca9825en.
76. Eggleston, H. S., Buendia, L., Miwa, K., Ngara, T. & Tanabe, K. *IPCC Guidelines for National Greenhouse Gas Inventories*. <http://www.ipcc-nggip.iges.or.jp/public/2006gl/index.htm> (2006).
77. Statistics Division of the FAO. *Guidelines on data collection for national statistics on forest products*. (2018).
78. Artés, T. *et al.* A global wildfire dataset for the analysis of fire regimes and fire behaviour. *Sci. Data* **6**, 296 (2019).
79. Giglio, L., Boschetti, L., Roy, D. P., Humber, M. L. & Justice, C. O. The Collection 6 MODIS burned area mapping algorithm and product. *Remote Sens. Environ.* **217**, 72–85 (2018).
80. Hansen, M. C. *et al.* High-Resolution Global Maps of 21st-Century Forest Cover Change. *Science* **342**, 850–853 (2013).

Lines 693-731 (revised manuscript track change) Describing the sensitivity analyses performed:

“The robustness of the model was assessed through five sensitivity analyses carried out on the most uncertain parameters and assumptions of the model: **(i)** temporal change in K parameter assumption and; **(ii)** changes in primary forest biomass density assumption; **(iii)** high fire impact assumption; **(iv)** low fire impact assumption; **(v)** effect of gross versus net area changes. As the CRAFT model reproduces the observed trends in forest biomass C stocks, its highest uncertainties lie in the estimation of the growth parameters. Here, two structural hypotheses of the model are likely to affect these calculations: **(i)** the assumption regarding the optimization of temporal changes in growth parameters and, **(ii)** the coefficient derived to estimate the temporal trend in primary forest biomass density. To assess the uncertainties associated to those assumptions, we ran 2 sensitivity analyses: **(i)** We tested an alternative hypothesis in which both the temporal trend of the r and K parameters were optimized against the reference assumption based on the optimization of the temporal change of the r parameter only **(ii)** We tested an alternative hypothesis in which biomass density in primary forest was taken as constant (using the benchmark provided in 2000 by Erb *et al.*¹²) against the reference assumption based on an estimation of changes in primary forest biomass density. . Burnt area is another source of uncertainty, because data are only available for the period after 2000 and for some countries missing altogether. Furthermore, it is intricate to separate burnt areas from deforestation areas. We thus consider the fire losses as the most uncertain input data to our model. Therefore, we carried out two additional sensitivity analyses: **(iii)** We tested the model with fire losses calculated by assuming the lowest fire severity coefficient (see table S3 and 4) and by assuming complete overlap between fires and deforestation in all countries (not only in tropical countries, as in the default estimation). **(iv)** We tested the model with fire losses calculated by assuming the highest fire severity coefficient (see table S4-5) assuming no overlap between forest area loss and burnt area in any countries. Last, as net area change may result in underestimating C fluxes^{81,82} we performed a sensitivity analysis **(v)** in which annual gross area gain and loss were estimated based on national conversion factor of gross to net area change derived from Li *et al.*²⁶ To do so, we used the data provided by Li *et al.*²⁶ to calculate the average net and average gross national forest area changes in 1992-2015. The ratio of net to gross area change was then derived by dividing the sum of the absolute value of all decreases and increases by the absolute value of the net change. This ratio was then used in the sensitivity analysis in order to quantify the effect of gross area change on forest biomass density (see Eq. 2). The differences in the model outputs and counterfactual scenarios assessments between the five sensitivity analyses and the default

model assumptions are presented in Table S2-3 and Fig. S5-S10”

References

12. Erb, K.-H. *et al.* Unexpectedly large impact of forest management and grazing on global vegetation biomass. *Nature* 553, 73–76 (2018).
26. Li, W. *et al.* Gross and net land cover changes in the main plant functional types derived from the annual ESA CCI land cover maps (1992–2015). *Earth Syst. Sci. Data* 10, 219–234 (2018).
81. Winkler, K., Fuchs, R., Rounsevell, M. & Herold, M. Global land use changes are four times greater than previously estimated. *Nat. Commun.* **12**, 2501 (2021).
82. Fuchs, R. *et al.* Assessing the influence of historic net and gross land changes on the carbon fluxes of Europe. *Glob. Change Biol.* **22**, 2526–2539 (2016).

Lines 248-272 (revised SII track change) Analyzing the sensitivity analysis results:

China switches from a net C sink driven by increase growth rate to a net C sink driven by reforestation in the gross area change sensitivity analysis (Fig. S6d). This is due to the fact that both area and growth rate changes have positive effects on the net C sink in China (Fig. S3) so that changes in model assumption could result in further highlighting one driver rather than the other.

India switches from a net C source driven by afforestation to a net C source driven by increased harvest pressure in the gross area change sensitivity analysis (Fig. S6d). Considering gross area changes in India resulted in such a decrease of biomass density (due to increase rejuvenation through higher reforested area) that harvest pressure became higher than annual NPP. In such conditions, the CRAFT model could not reproduce the FRA data in Indian managed forest, which even collapsed by the end of the simulation in this sensitivity analysis, thus suggesting that the average gross-net area change ratio of c.12 in India³ might be unrealistic.

Australia switches from a net C sink driven by deforestation to a net C sink driven by increased growth rate in the dynamic K and low fire sensitivity analyses (Fig. S6b and f) but also to a net C source driven by increased fire intensity in the constant primary biomass and high fire sensitivity analyses (Fig. S6c and e). These changes in the type of C-dynamics trajectory identified in Australia are due to the fact that several drivers had large but contradicting effects in these countries: Both increased fire intensity and deforestation contributed to a net C source while increased growth rate was actually the largest driver counteracting the C source in the reference model simulation (Fig. S3). In those conditions, it is unsurprising that changes in the model assumption and parameters are more likely to affect the C budget and main driver attribution in this country.

In addition, it is worth noting that many West European countries switch from a net C sink driven by increased growth rate to a net C sink driven by reforestation and *vice-versa* across the different sensitivity analyses. These changes result from the synergetic effects of both reforestation and growth rate changes in these countries (Fig. S3), so that changes in model assumption may result in further highlighting one driver rather than the other.

Reference

26. Li, W. *et al.* Gross and net land cover changes in the main plant functional types derived from the annual ESA CCI land cover maps (1992–2015). *Earth Syst. Sci. Data* 10, 219–234 (2018).

Q.1.4. Finally, the fire emissions estimates using the global coefficients are grossly over estimated, especially for coniferous forests 1-4. These need to be much lower given that most of the biomass in a forest is in live trees, and they really don't combust (~5% combustion).

A.1.4. We thank the reviewer for drawing our attention on these papers. After carefully reading them, we agree on the fact that combustion factors derived from plot observation are close to 5%. Following the reviewer remark we further dug into the literature and figured out that we previously did not correctly employ the fuel load and combustion efficiency coefficients and that we erroneously considered coefficients reported for litter and dead wood while our focus is here live biomass only. Consequently, we have re-calculated the fire losses component in the model parameter with coefficients of overall fire severity now ranging from 4.2 to 10.7%. This modification in the fire losses estimates implied to entirely re-run the model and re-do all figures (as well as the sensitivity analysis), which also explains the slight differences between the results displayed in the revised figures of the main manuscript versus in the previous ones. This modification changed the overall result, since the estimate of global C emissions in 1990-2020 amounts to 0.74 GtC in the revised manuscript versus 2.4 GtC in the original manuscript, and down-graded the role of fire. However, it did not affect our other conclusions, in particular not the main finding on the pivotal role of changes in growth conditions and changes in forest areas.

The new calculation procedure of fire losses has been corrected in the method section as follows:

Lines 563-589 (revised manuscript track change)

“Data on forest area burnt were extracted from the last FRA (<https://fra-data.fao.org/>). From this data, we calculated the annual live tree biomass loss by fire such as:

$$FL = (C_w \times FL_w \times \alpha_w + C_l \times FL_l \times \alpha_l) \times (A_b/A) \times B \quad [\text{Eq. 3}]$$

With FL the annual loss by fire (tC/yr); C the fraction of the respective tree compartment (w = wood, i.e. branches and stem; l = leaves); FL the respective fuel load i.e. the % of live biomass that can actually burn per tree compartment; α the combustion efficiency coefficient, i.e., the % of fuel load consumed by fire per tree compartment in an area affected by fire; (A_b/A) , the fraction of area burnt over the total forest area and B the standing biomass (tC). We assumed that the fraction of area burnt was the same in managed and primary forest as in total country-level forest.

Since our study is focused on C in living tree biomass alone, we used stem and foliage as a proxy for living tree biomass, instead of calculating total forest fuels i.e. understory, litter, duff, deadwood (Table S6). We used the assessment by Hoelzemann et al.⁶⁶, which provides the percentage of total tree biomass compartment that is available for combustion (i.e. fuel load) per biome (tropical, temperate, boreal Eurasian, American forest). We assessed a range of studies reporting combustion efficiency coefficients for these biomes based on fire models and field observations⁶⁷⁻⁷⁰ to derive plausible values for combustion efficiency per tree compartment (see Table S7). Consequently, we calculated three fire severity levels (low, moderate, high), hence assessing a range of coefficients (*fire severity* (%) = $\sum_i C_i \times FL_i \times \alpha_i$), which were used to perform sensitivity analyses (see below). We used the moderate fire severity coefficients (ranging from 4 to 11% across world countries) to run the reference model simulations (figures

displayed in the main manuscript).”

References

66. Hoelzemann, J. J. Global Wildland Fire Emission Model (GWEM): Evaluating the use of global area burnt satellite data. *J. Geophys. Res.* **109**, D14S04 (2004).
67. Mouillot, F., Narasimha, A., Balkanski, Y., Lamarque, J.-F. & Field, C. B. Global carbon emissions from biomass burning in the 20th century: GLOBAL CARBON EMISSIONS FROM BIOMASS BURNING. *Geophys. Res. Lett.* **33**, n/a-n/a (2006).
68. Yang, J. *et al.* A growing importance of large fires in conterminous United States during 1984–2012. *J. Geophys. Res. Biogeosciences* **120**, 2625–2640 (2015).
69. Kloster, S. *et al.* Fire dynamics during the 20th century simulated by the Community Land Model. *Biogeosciences* **7**, 1877–1902 (2010).
70. van Leeuwen, T. T. *et al.* Biomass burning fuel consumption rates: a field measurement database. *Biogeosciences* **11**, 7305–7329 (2014).

Q.1.5. There should probably be some references to wood product carbon emissions 5, 6 and substitution benefits 7 in some sections to support the conclusions.

A.1.5. We thank the reviewers for the literature suggestions which have been useful to further reinforce the message delivered in the last paragraph.

Lines 563-589 (revised manuscript track change, *also quoted above in A.2.2*)

“In this context, increasing forest harvest volumes – a strategy often promoted in the course of climate change-mitigation efforts embraced as the “bioeconomy” – , appears to have critical unintended side-effects, despite the potential of wood for substituting some emissions-intensive products and processes^{53–55}: By not only reducing the carbon sink function in forests, but also accelerating the overall C turnover rates through rejuvenation of forests and transfer to harvested wood products of lifetimes shorter than those of old-growth forests^{56–58}, such strategies result in a critical loss of C sink capacity.”

53. Law, B. E. *et al.* Land use strategies to mitigate climate change in carbon dense temperate forests. *Proc. Natl. Acad. Sci.* **115**, 3663–3668 (2018).
54. Braun, M. *et al.* A holistic assessment of greenhouse gas dynamics from forests to the effects of wood products use in Austria. *Carbon Manag.* **7**, 271–283 (2016).
55. Harmon, M. E. Have product substitution carbon benefits been overestimated? A sensitivity analysis of key assumptions. *Environ. Res. Lett.* **14**, 065008 (2019).
56. Erb, K.-H. *et al.* Biomass turnover time in terrestrial ecosystems halved by land use. *Nat. Geosci.* **9**, 674–678 (2016).
57. Hudiburg, T. W., Law, B. E., Moomaw, W. R., Harmon, M. E. & Stenzel, J. E. Meeting GHG reduction targets requires accounting for all forest sector emissions. *Environ. Res. Lett.* **14**, 095005 (2019).
58. Dymond, C. C. Forest carbon in North America: annual storage and emissions from British Columbia’s harvest, 1965–2065. *Carbon Balance Manag.* **7**, 8 (2012).

Other:

Q.1.6. Line 10: "approached" should be "approach"

A.1.6. Corrected thanks

Q.1.7. Line 82: there is a word missing ("us?")

A.1.7. Corrected thanks

Q.1.8. Line 85/86: This is a very clear finding and it is unsurprising (Deforestation is driving emissions).

Shouldn't this be in the abstract more clearly?

A.1.8. We have modified the abstract so that the main results are now presented in a more impactful way. Please see our previous answer.

Q.1.9. Lines 178 to 180 also seem really important. Primary forest is not the reason for 'non-enhanced growth', etc. but this was not mentioned until this point.

A.1.9. Thank you for this remark, we have now inserted the following sentence at the end of the first section following the Introduction:

Lines 96-101 (revised manuscript track change, also quoted above in A.2.3)

“In addition, our finding that the forest growth rate increased annually by 0.19, 0.21 and 0.21 % from 1990 to 2020 respectively for primary, managed and total forests of the world is consistent with Kolby Smith et al.²⁰ who find that also Net Primary Production (NPP) increased annually between 0.10-0.25% in the period 1982-2011, as well as with other modelling and remote-sensing studies documenting a global greening trend, i.e. vegetation thickening following increased vegetation growth rate^{21,22}.”

References

20. Kolby Smith, W. *et al.* Large divergence of satellite and Earth system model estimates of global terrestrial CO₂ fertilization. *Nat. Clim. Change* **6**, 306–310 (2016).
21. Zhu, Z. *et al.* Greening of the Earth and its drivers. *Nat. Clim. Change* **6**, 791–795 (2016).
22. Zhao, L., Dai, A. & Dong, B. Changes in global vegetation activity and its driving factors during 1982–2013. *Agric. For. Meteorol.* **249**, 198–209 (2018).

Q.1.10. Lines 207 onwards: Title and abstract do not reflect this. Many will interpret enhanced growth as 'younger forests grow faster so we need to harvest the old ones and plant young ones'.

A.1.10. We have changed the title and abstract following the reviewer remarks (please see our answers above).

Q.1.11. Suggested refs:

1. Stenzel, J.E., K.J. Bartowitz, M.D. Hartman, J.A. Lutz, C.A. Kolden, A.M.S. Smith, . . . T.W. Hudiburg, Fixing a snag in carbon emissions estimates from wildfires. *Global Change Biology*, 2019. 0(0). DOI: 10.1111/gcb.14716.
2. Hudiburg, T.W., B.E. Law, W.R. Moomaw, M.E. Harmon, and J.E. Stenzel, Meeting

- GHG reduction targets requires accounting for all forest sector emissions. *Environmental Research Letters*, 2019. 14(9): p. 095005. DOI: 10.1088/1748-9326/ab28bb.
3. Campbell, J., D. Donato, D. Azuma, and B. Law, Pyrogenic carbon emission from a large wildfire in Oregon, United States. *Journal of Geophysical Research: Biogeosciences*, 2007. 112(G4): p. G04014. DOI: 10.1029/2007JG000451.
 4. Ghimire, B., C.A. Williams, G.J. Collatz, and M. Vanderhoof, Fire-induced carbon emissions and regrowth uptake in western U.S. forests: Documenting variation across forest types, fire severity, and climate regions. *Journal of Geophysical Research: Biogeosciences*, 2012. 117(G3). DOI: doi:10.1029/2011JG001935.
 5. Hudiburg, T.W., B.E. Law, W.R. Moomaw, M.E. Harmon, and J.E. Stenzel, Meeting regional GHG reduction targets requires accounting for all forest sector emissions. *Environmental Research Letters*, 2019. 14(9): p. 095005.
 6. Dymond, C.C., Forest carbon in North America: annual storage and emissions from British Columbia's harvest, 1965–2065. *Carbon Balance and Management*, 2012. 7: p. 8. DOI: 10.1186/1750-0680-7-8.
 7. Harmon, M.E., Have product substitution carbon benefits been overestimated? A sensitivity analysis of key assumptions. *Environmental Research Letters*, 2019. 14(6): p. 065008.

A.1.11. We are very grateful for the suggestions of these additional references which have been very useful for improving the estimates of fire efficiency coefficient used in our modelling approach and revised the section on fire activities in the SI.

Reviewer #2 (Remarks to the Author):

***Q.2.1.* Enhanced growth – more than reforestation – counteracted biomass carbon emissions (1990-2020), by Julia Le Noë et al.**

The manuscript by Le Noë et al. assesses drivers of changes in global forest biomass for 1990-2020 using an approach that simulates biomass changes at country level. For this, national statistics from FAO's Forest Resource Assessment are used in a simple model setup representing the main processes and drivers of forest biomass change.

The study addresses an important question with relevance to the broad scientific audience, and the topic fits well with the strong focus on forest biomass change as a means of climate change mitigation. The use of FAO's national data in the study is a welcome supplement to the literature, which is at these scales otherwise dominated by data sets based on gridded inventory data, remote sensing-derived estimates and (gridded) vegetation modelling.

I have some questions about the methods, and in particular about the implications of the use of these FAO statistical data at country level. Apart from that, I am not very convinced about the typology used, as it may promote drivers that are not the largest single effects in case two or more other drivers oppose this change (details below).

Having said that, I think the study is a very welcome contribution to the ongoing discussions about changes in the forest carbon sink and I warmly recommend publication of this study once my concerns below are addressed.

I provide detailed comments below to aid revisions of the manuscript.

***A.2.1.* We thank the reviewer for the constructive remarks and appreciation of our manuscript. We have done our best to reply to all comments addressed by the reviewer (see our reply below).**

In addition, reviewer 1's remark on fire loss estimates led us to reconsider the parameter we originally used for fire severity. Consequently, we have re-calculated the fire losses component in

the model parameter with coefficients of overall fire severity now ranging from 4.2 to 10.7%. This modification in the fire losses estimates implied to entirely re-run the model and re-do all figures (as well as the sensitivity analysis). This modification changed the overall result, since the estimate of global C emissions in 1990-2020 amounts to 0.74 GtC in the revised manuscript versus 2.4 GtC in the original manuscript, and down-graded the role of fire. However, it did not affect our other conclusions, in particular not the main finding on the pivotal role of changes in growth conditions and changes in forest areas.

Furthermore, this comment made us reconsider and improve the typology (see below).

Q.2.2. I think that the fact that drivers (in the counterfactual simulations) are assessed as aggregated to national level should be better reflected in the discussion. E.g., when showing totals disaggregated into increase and decrease (e.g., Fig. 2ab), these increase and decrease are based on national numbers, whereas there also may be a composite of increase and decrease within each country.

A.2.2. We thank the reviewer for this important remark regarding the scale effect on the model robustness and results reliability. To address this remark, we revised the manuscript and SI in two ways:

- 1- In order to assess the robustness of our modelling approach, we carried out 5 sensitivity analyses on the most uncertain parameters/assumptions, ***including the effect of country-level data aggregation***: (i) temporal change in K parameter assumption; (ii) constant primary forest biomass density assumption; (iii) low fire impact assumption; (iv) high fire impact assumption; (v) effect of gross versus net area changes. These sensitivity analyses allow us to provide error bars where relevant in the manuscript figures (Fig. 2a & b, Fig. 3c). The main figures arising from these five sensitivity analyses are also displayed in the Supplementary Information (see Tables S2-3 and Fig. S5-10). Overall the comparison between the results presented in the main manuscript *versus* those displayed in the SI as part of the sensitivity analysis confirmed the robustness of the model and the reliability of our main conclusions: there were only differences in the magnitudes of the estimates by the various sensitivity tests but the results from the sensitivity analyses did not affect our main conclusion that changes in growth conditions more than reforestation counteracted the biomass C emissions mainly caused by deforestation in 1990-2020.
- 2- We have introduced several paragraphs to better reflect the implications of our national level assessment. In addition, we discuss the model uncertainties, put our results and conclusions in context with other approaches and carried out a literature survey in order to provide independent comparisons between our modelled estimates/FRA dataset of C budgets and estimates by other independent studies (Table S1): We find that our results and conclusions are corroborated by other researches. When comparing our results with other studies, some using net other gross area changes, we are also in good agreement with all other independent estimates. We take this as further indications that our approach is valid. However, we also point that contrary to existing study, our simple modelling approach can be run with parsimonious data availability and allows to disentangle the major drivers behind forest C stock and flux trajectories.

We revised the manuscript accordingly. The additional paragraphs inserted in the main

manuscript read as follows:

Lines 64-70 (revised manuscript track change, see also lines 693-731 quoted below)

Commenting the sensitivity assessment:

“The five sensitivity analyses carried out on the most uncertain model inputs and assumptions (see method descriptions and Fig.S5-10) confirmed the results presented in Fig.1: The largest deviation derived from the sensitivity analysis considering forest gross instead of net area changes results in a relative RSME of 1.94% with global C emissions c.2.5 times higher than the FRA estimates (Table S2). The simulations from the reference model assumptions yield the best RMSE and closest agreement with the C budgets derived from the FRA, indicating that they are the most optimal.”

Lines 76-86 (revised manuscript track change) *Putting our results in context with other independent estimates (see also Table S1):*

“These figures are within the range of the estimated sink in forest soil and biomass of 0.1 ± 7.3 GtC/yr in 2001-2019 found by Harris et al¹⁷. Our estimation is also consistent with that of Tubiello et al³ of 0.11 GtC/yr net C emissions from forest ecosystems. A comparison³ of FRA-derived global forest C emissions with other independent estimates reported in 1990-2015 by National Greenhouse Gas Inventories (NGHGs) – including the Russian Federation, the USA, China, Indonesia and India – and by the United Nations Framework Convention on Climate Change for other countries (UNFCCC, 2020¹⁸) yields a slight difference of c.18%, although the UNFCCC and NGHGI’s account, by definition, only for emissions from managed land³. Further independent comparisons at the national and macro-regional levels are compiled in Table S1 and reveal that C emissions estimated in the present study are in good agreements with other researches.”

References

3. Tubiello, F. N. *et al.* Carbon emissions and removals from forests: new estimates, 1990–2020. *Earth Syst. Sci. Data* **13**, 1681–1691 (2021).
17. Harris, N. L. *et al.* Global maps of twenty-first century forest carbon fluxes. *Nat. Clim. Change* (2021) doi:10.1038/s41558-020-00976-6.
18. UNFCCC: United Nations Framework Convention on Climate Change: *Time Series – Annex I, UNFCCC*. https://di.unfccc.int/time_series (Bonn).

Lines 95-104 (revised manuscript track change) *Putting our main finding in context with other independent research:*

“This is in line with other research pointing to the relevance of biomass thickening for forest C sequestration¹⁹. In addition, our finding that the forest growth rate increased annually by 0.19, 0.21 and 0.21 % from 1990 to 2020 respectively for primary, managed and total forests of the world is consistent with Kolby Smith et al.²⁰ who find that also Net Primary Production (NPP) increased annually between 0.10-0.25% in the period 1982-2011, as well as with other modelling and remote-sensing studies documenting a global greening trend, i.e. vegetation thickening following increased vegetation growth rate^{21,22}. Note that estimates of annual growth rate increase in 1990-2020 by the sensitivity analyses provide narrow ranges of 0.17-0.19, 0.21-0.23

and 0.20-0.22% respectively for primary, managed and total forests of the world (Table S2).”

References

19. Kauppi, P. E. *et al.* Carbon benefits from Forest Transitions promoting biomass expansions and thickening. *Glob. Change Biol.* **26**, 5365–5370 (2020).
20. Kolby Smith, W. *et al.* Large divergence of satellite and Earth system model estimates of global terrestrial CO₂ fertilization. *Nat. Clim. Change* **6**, 306–310 (2016).
21. Zhu, Z. *et al.* Greening of the Earth and its drivers. *Nat. Clim. Change* **6**, 791–795 (2016).
22. Zhao, L., Dai, A. & Dong, B. Changes in global vegetation activity and its driving factors during 1982–2013. *Agric. For. Meteorol.* **249**, 198–209 (2018).

Lines 138-153 (revised manuscript track change) Commenting the sensitivity assessment regarding the effect of gross versus net area change:

“A sensitivity analysis on the potential underestimation of C dynamics resulting from the use of net area change data at country level (see methods section and Fig. S5) reveals that accounting for gross area changes²⁶ instead of net area change would result in higher global C emissions estimates (4.19 GtC in the sensitivity test versus 0.74 GtC in the reference simulation) but would reveal the same patterns of forest C-dynamic drivers (Fig. S5). However, the magnitude of the main drivers would be slightly changed with a lower effect of changes in area (C sink in the hypothetical absence of area changes reaching 20.8 GtC in the sensitivity tests versus 26.9 GtC in the reference assessment) and a higher effect of growth rate changes (C source in the hypothetical absence of growth rate changes reaching 13.1 GtC in the sensitivity tests versus 7.4 GtC in the reference assessment). Generally, the range of results derived from the five sensitivity analyses does not change the relative importance of the individual drivers in any of the scenarios (Table S3, Fig. 2a and S5). However, the sensitivity analyses highlight that the uncertainty is large enough to reverse the cumulated C signal in the absence of changes in harvest (CF 1), changes in burnt area (CF 3), and the complete absence of burnt areas (CF 6). By contrast, the signals of CF 2 (no growth rate change), CF 4 (no area change) and CF 5 (no harvest) are larger than the uncertainty across sensitivity analyses, signaling that our findings on these drivers are most robust.”

Reference

26. Li, W. *et al.* Gross and net land cover changes in the main plant functional types derived from the annual ESA CCI land cover maps (1992–2015). *Earth Syst. Sci. Data* **10**, 219–234 (2018).

Lines 167-180 (revised manuscript track change) Reflecting on the implication of using net versus gross area change data

“The fact that we use here country-level data comes both with limitations and advantages. The main limitation associated with national data is that it conceals gross C fluxes in forest biomass dynamics and blurs heterogeneity in growth conditions and anthropogenic management within countries. The country-level resolution aggregates the effects of manifold, partly counteracting processes at the local level – including photosynthesis, maintenance respiration, growth respiration, as well as forest area loss and expansion – on the annual dynamic of primary and managed forest biomass. As a consequence, our optimization of the growth function actually reflects apparent national growth rates resulting from the aggregate of these processes. However,

this simplification of forest ecosystem functioning is also an advantage. Our approach reproduces forest biomass dynamics very accurately, which is complementary to most process-based models aimed at depicting biological processes and their abiotic controls²⁷ but providing a wide range of C flux estimations¹ and hardly reproducing observation from inventory data^{1,28,29}. By contrast, the strength of the modelling approach implemented here is that it can be run with parsimonious data availability and allows to disentangle the major drivers behind forest C stock and flux trajectories.”

References

1. Friedlingstein, P. *et al.* Global Carbon Budget 2020. *Earth Syst. Sci. Data* **12**, 3269–3340 (2020).
27. Pongratz, J. *et al.* Models meet data: Challenges and opportunities in implementing land management in Earth system models. *Glob. Change Biol.* **24**, 1470–1487 (2018).
28. Bellassen, V. *et al.* Reconstruction and attribution of the carbon sink of European forests between 1950 and 2000. *Glob. Change Biol.* **17**, 3274–3292 (2011).
29. Ciais, P. *et al.* Carbon accumulation in European forests. *Nat. Geosci.* **1**, 425–429 (2008).

Lines 231-246 (revised manuscript track change) Commenting the sensitivity assessment:

“These highlights derived from the typology remained the same in all sensitivity analyses (Fig. S6-7), despite some possible changes in country type identification (Fig. 3a and S6) and amplitude shifts in the attribution of main drivers globally (Fig. 3c and S7). The ranges of values in the attribution of main drivers result from the previously reported differences between the counterfactual and actual C budget estimates across sensitivity analyses (see also Tables S2-3) combined with some changes in the type of forest C-dynamics trajectory identified through the typology in countries with large forest biomass stocks: China, India and Australia (Fig. S6 and explanations below Fig. S6). However, these shifts do not affect the main conclusions derived from Fig. 3c: In all sensitivity analyses, growth rate changes remain the main driver of global forest biomass C sink with total net C sinks in countries where increasing growth rate is the main driver (including both solid and hatched countries) ranging from 12.1 to 21.1 GtC, while afforestation always holds the second place of global C sink driver (total net C sinks in countries where afforestation is the main driver ranging from 2.4 to 7.7 GtC). Similarly, total net C sources by countries where deforestation is the main driver range from -21.9 to -14.0 GtC, thus highlighting that deforestation would by far remain the main driver of forest biomass C emissions across all sensitivity analyses.”

Lines 572-578 (revised manuscript track change) Discussing data quality:

“According to the last FRA report², 86% of the forest area reported was estimated following an IPCC⁷⁶ tier 3 approach and 88% of the growing stock trend following tiers 2 or 3 approaches. Similarly, the harvest data reported by the FAO statistics are subject to several statistical tests, data validation and reconciliation with other data sources⁷⁷. The data reported by the FRA for burnt area arise from the combination of several remote-sensing approaches including the Global Wildfire Information System⁷⁸, the Moderate-Resolution Imaging Spectroradiometer (MODIS)⁷⁹ and the Global Forest Change product⁸⁰.”

References

2. *Global Forest Resources Assessment 2020*. (FAO, 2020). doi:10.4060/ca9825en.
76. Eggleston, H. S., Buendia, L., Miwa, K., Ngara, T. & Tanabe, K. *IPCC Guidelines for National Greenhouse Gas Inventories*. <http://www.ipcc-nggip.iges.or.jp/public/2006gl/index.htm> (2006).
77. Statistics Division of the FAO. *Guidelines on data collection for national statistics on forest products*. (2018).
78. Artés, T. *et al.* A global wildfire dataset for the analysis of fire regimes and fire behaviour. *Sci. Data* **6**, 296 (2019).
79. Giglio, L., Boschetti, L., Roy, D. P., Humber, M. L. & Justice, C. O. The Collection 6 MODIS burned area mapping algorithm and product. *Remote Sens. Environ.* **217**, 72–85 (2018).
80. Hansen, M. C. *et al.* High-Resolution Global Maps of 21st-Century Forest Cover Change. *Science* **342**, 850–853 (2013).

Lines 693-731 (revised manuscript track change *Describing sensitivity assessment*):

“The robustness of the model was assessed through five sensitivity analyses carried out on the most uncertain parameters and assumptions of the model: **(i)** temporal change in K parameter assumption and; **(ii)** changes in primary forest biomass density assumption; **(iii)** high fire impact assumption; **(iv)** low fire impact assumption; **(v)** effect of gross versus net area changes. As the CRAFT model reproduces the observed trends in forest biomass C stocks, its highest uncertainties lie in the estimation of the growth parameters. Here, two structural hypotheses of the model are likely to affect these calculations: **(i)** the assumption regarding the optimization of temporal changes in growth parameters and, **(ii)** the coefficient derived to estimate the temporal trend in primary forest biomass density. To assess the uncertainties associated to those assumptions, we ran 2 sensitivity analyses: **(i)** We tested an alternative hypothesis in which both the temporal trend of the r and K parameters were optimized against the reference assumption based on the optimization of the temporal change of the r parameter only **(ii)** We tested an alternative hypothesis in which biomass density in primary forest was taken as constant (using the benchmark provided in 2000 by Erb *et al.*¹²) against the reference assumption based on an estimation of changes in primary forest biomass density. . Burnt area is another source of uncertainty, because data are only available for the period after 2000 and for some countries missing altogether. Furthermore, it is intricate to separate burnt areas from deforestation areas. We thus consider the fire losses as the most uncertain input data to our model. Therefore, we carried out two additional sensitivity analyses: **(iii)** We tested the model with fire losses calculated by assuming the lowest fire severity coefficient (see table S3 and 4) and by assuming complete overlap between fires and deforestation in all countries (not only in tropical countries, as in the default estimation). **(iv)** We tested the model with fire losses calculated by assuming the highest fire severity coefficient (see table S4-5) assuming no overlap between forest area loss and burnt area in any countries. Last, as net area change may result in underestimating C fluxes^{81,82} we performed a sensitivity analysis **(v)** in which annual gross area gain and loss were estimated based on national conversion factor of gross to net area change derived from Li *et al.*²⁶ To do so, we used the data provided by Li *et al.*²⁶ to calculate the average net and average gross national forest area changes in 1992-2015. The ratio of net to gross area change was then derived by dividing the sum of the absolute value of all decreases and increases by the absolute value of the net change. This ratio was then used in the sensitivity analysis in order to quantify the effect of gross area change on forest biomass density (see Eq. 2). The differences in the model outputs and counterfactual scenarios assessments between the five sensitivity analyses and the default

model assumptions are presented in Table S2-3 and Fig. S5-S10”

References

12. Erb, K.-H. *et al.* Unexpectedly large impact of forest management and grazing on global vegetation biomass. *Nature* 553, 73–76 (2018).
26. Li, W. *et al.* Gross and net land cover changes in the main plant functional types derived from the annual ESA CCI land cover maps (1992–2015). *Earth Syst. Sci. Data* 10, 219–234 (2018).
81. Winkler, K., Fuchs, R., Rounsevell, M. & Herold, M. Global land use changes are four times greater than previously estimated. *Nat. Commun.* **12**, 2501 (2021).
82. Fuchs, R. *et al.* Assessing the influence of historic net and gross land changes on the carbon fluxes of Europe. *Glob. Change Biol.* **22**, 2526–2539 (2016).

Lines 248-272 (revised SII track change) Analyzing the results of sensitivity analyses

“China switches from a net C sink driven by increase growth rate to a net C sink driven by reforestation in the gross area change sensitivity analysis (Fig. S6d). This is due to the fact that both area and growth rate changes have positive effects on the net C sink in China (Fig. S3) so that changes in model assumption could result in further highlighting one driver rather than the other.

India switches from a net C source driven by afforestation to a net C source driven by increased harvest pressure in the gross area change sensitivity analysis (Fig. S6d). Considering gross area changes in India resulted in such a decrease of biomass density (due to increase rejuvenation through higher reforested area) that harvest pressure became higher than annual NPP. In such conditions, the CRAFT model could not reproduce the FRA data in Indian managed forest, which even collapsed by the end of the simulation in this sensitivity analysis, thus suggesting that the average gross-net area change ratio of c.12 in India³ might be unrealistic.

Australia switches from a net C sink driven by deforestation to a net C sink driven by increased growth rate in the dynamic K and low fire sensitivity analyses (Fig. S6b and f) but also to a net C source driven by increased fire intensity in the constant primary biomass and high fire sensitivity analyses (Fig. S6c and e). These changes in the type of C-dynamics trajectory identified in Australia are due to the fact that several drivers had large but contradicting effects in these countries: Both increased fire intensity and deforestation contributed to a net C source while increased growth rate was actually the largest driver counteracting the C source in the reference model simulation (Fig. S3). In those conditions, it is unsurprising that changes in the model assumption and parameters are more likely to affect the C budget and main driver attribution in this country.

In addition, it is worth noting that many West European countries switch from a net C sink driven by increased growth rate to a net C sink driven by reforestation and *vice-versa* across the different sensitivity analyses. These changes result from the synergetic effects of both reforestation and growth rate changes in these countries (Fig. S3), so that changes in model assumption may result in further highlighting one driver rather than the other.”

Q.2.3. Although I see the importance of some sort of typology to summarize main drivers, I am in doubt whether the current scheme (supplementary material S3) is accurate. By separating early on in the scheme between increase and decrease, one can easily end up in a situation where individual factors with the largest change do not show up, e.g. with a large

increase or decrease in area counterbalanced by two other factors (e.g. changes in growth or in harvest) – if the sum of the latter exceeds that of the former, it is one of the latter factors that will be highlighted.

4.2.3. We thank the reviewer for this remark which helped us to improve the typology. Indeed, the typology is needed in the paper to be able to summarize the findings in a succinct way. In the previous version, the decision tree may indeed, in some cases, result in concealing one major driver if the effect of this driver on biomass C stocks was overcompensated by the combined effect of two or more other drivers. For instance, if in a country there were a C sink with growth change being the main driver of the sink (according to the typology based on the CF assessment), then this country would be identified as belonging to the type ‘C sink driven by enhanced growth’. However, if in this country change in area actually had the strongest absolute effect (according to the CF assessment) on forest biomass dynamics, but this effect counteracted the actual C budget, then the typology identification would actually conceal this major driver. Therefore, in order to better account for possible antagonistic effects, we have now introduced a new trajectory type reflecting cases in which the strongest driver counteracted the observed C budget trend, but apparently was overcompensated for.

We revised the manuscript by: (i) modifying the decision tree and presenting it in the main manuscript (see Fig. 3b); (ii) inserting new sentences/paragraphs in the ‘Typology of forest biomass change’ section:

Lines 200-205 (revised manuscript track change)

“However, as the early separation between increasing and decreasing biomass C stocks in the decision tree (Fig. 3b) may conceal the effect of a major driver counteracting the observed C dynamic, the typology also accounts for possible antagonistic effects by identifying cases in which the main driver of observed C dynamics is not, in absolute terms, the most important driver (e.g., C stocks increase but the driver with the strongest absolute effect counteracts this positive budget, see also Fig. S3).”

Lines 210-213 (revised manuscript track change)

“The net C emissions by countries where deforestation is the most significant driver reach c. 21.3 GtC, with only 0.3 GtC of these emissions being counteracted by another major driver (either increased growth rate or lower harvest pressure). These emissions represent c. 92.7% of the 21.9 GtC net emissions arising from all countries acting as net C sources (Fig. 3c).”

Lines 215-220 (revised manuscript track change)

“The net C sinks by countries where changes in forest growth rates are the main driver reach c. 16.4 GtC, with 0.9 GtC of these sinks being counteracted by another major driver (increased harvest pressure in all cases except for Sudan where area loss was the major driver counteracting the C sink). These C sinks mainly driven by increased growth rate represented c. 77.5% of the 21.1 GtC net sink created by all countries acting as net C sinks (Fig. 3c).”

Lines 221-225 (revised manuscript track change)

“Forest area expansion from 1990 to 2020 is the main driver of forest biomass net C sink in only a few Northern countries but also some Southern countries, namely Vietnam, India and Chile, in

line with findings reported for these countries³²⁻³⁴, all together accounting for a net C sink of 3.9 GtC. However, more than half of the C sinks mainly driven by reforestation are counteracted by another major driver (either declining forest biomass growth rate or increased harvest pressure).”

Q.2.4. Also, I would like to suggest moving the explanation of the typology (or a short description of it) to the main text. It is key to understand Fig. 3, but at the moment “hidden” in the supplementary material.

A.2.4. We have now integrated the decision tree of the typology to Fig. 3 in the main manuscript (see revised figure and caption)

“On the one hand, several studies highlight the feedback of environmental variables...”: The effects mentioned here are drivers, not feedbacks.

A.2.5. We have changed the sentence as follows: “On the one hand, several studies highlight the effects of environmental drivers...” (lines 276-277 in the revised manuscript track change).

Q.2.6. Fig. 4 caption: This is not a Marimekko diagram – it lacks the spatial (“Marimekko”) pattern of it. But it is a nice attempt to visualize the data. Does it contain all 152 countries? The number of points seems limited, but this may also be because data are rounded to whole percent numbers. A slightly larger version of the figure would help with the annotation of countries (which I think is very informative here). And you could consider to draw the rectangles between the line and the x-axis to highlight area changed. You could also decide to use colours again in your annotation to highlight the importance of the different drivers.

A.2.6. Thanks for your appreciation. Yes, the graphs contain all 152 countries except for primary forests where only the countries with primary forests are plotted. Following this comment, we have increased the size of the figure and we have used the same colors for dots plotted as the ones presented in the typology in Fig. 3. However, we preferred to stick with the initial diagram style and therefore we do not call it a ‘Marimekko diagram’ anymore, but simply a ‘diagram’.

Q.2.7. The method, and with that the outcome of the study, depend greatly on the use of the FRA data, and on the assumptions made for that. I would welcome a more thorough discussion of the results and the uncertainties that are introduced by these data. Some questions and thoughts that could be relevant for that:

- The aggregation of data to country level makes that in some cases greatly different areas/types of forest are lumped together. Are the assumed uniform changes within the country (one growth rate, one total biomass, one rate of afforestation/deforestation, etc.) accurate?

- Eq. 2: In the last part, $\text{MIN}(1, A(n)/A(n+1))$ seems to indicate that biomass density is assumed to decrease with increasing area (to account for the establishment of new forest with low initial biomass, I presume). Does the growth function (Eq. 1) capture the biomass increase accurately when doing so? This assumes basically that the existing biomass is “diluted” over a larger area, but does this give a good average of the unequal growth between the “old”, established area and the added area with more rapid growth of new forest?

- How do the incomplete data on forest fires affect the estimated role of fire? What is the risk that this is now erroneously accounted for in e.g. altered linear growth trends?

A.2.7. The reviewer is perfectly right about Eq. 2 and its implication on biomass density ‘dilution’ in case of forest area expansion. This calculation provides a good average of the unequal growth between the “old” and the “young” forest growing on afforested area because the logistic function between NPP and biomass is originally derived from Yield Table data with uneven tree age distribution (see SOM3 in Le Noë et al., 2020). This is actually one of the main advantages of the CRAFT model: the simple relationship between NPP and B allows to circumvent the difficulties and uncertainties associated with the reconstruction of the age structure.

Le Noë et al. (2020) <https://doi.org/10.1111/gcb.15004>

We answered the other comments and suggestions made by the reviewer in our previous answer A.2.2. as we now:

- Put our estimates in context with other studies, including estimates from process-based approach (see lines 76-86 in the revised manuscript track change, Table S1 in the revised SI1 track change, *also quoted in A.2.2*).
- Further confronted our main findings – changes in growth conditions are key driver counteracting forest C emissions – with other modelling studies (see lines 95-104 in the revised manuscript track change, *also quoted in A.2.2*).
- Added reflection on the scale issue (see lines 167-180 in the revised manuscript track change, *also quoted in A.2.2*).
- Performed five sensitivity analyses (including the effect of gross/net area change, high and low estimates of fire losses) which allowed us to better evaluate the robustness of our modelling approach and to provide error bar where relevant (see lines 64-70, 138-153, 231-246, 693-731 in the revised manuscript track change, Tables S2-3, Fig. S5-10 and lines 248-272 in the revised SI track change, *also quoted in A.2.2*)
- Extended the discussion on the uncertainty related to the FRA and FAO data from which we derived most of the input data for our modelling approach (see lines 572-578 in the revised manuscript track change, *also quoted in A.2.2*).

Q.2.8. Input data forest area: From the given description, “production forest” contains both planted and naturally regenerated forest (secondary forest). How large are these two contributions to the total of “production forest”? If the secondary forest is a significant contributor, it may need a more inclusive title, as secondary forest is not necessarily a production forest.

Thanks for this remark, we have now changed ‘production forest’ for ‘managed forest’. Following the FAO definition, this includes both ‘planted forest’ and, ‘naturally regenerated forest’. In addition, as FAO reported ‘mangrove’ which can be both primary or managed forest, we assumed half of the ‘mangrove’ as managed forest (the rest being primary). As the mangrove area is generally very small in comparison to other forest definition, this assumption can only have very minor influence on our simulations.

Reviewer #3 (Remarks to Author):

Generally the article is interesting, the finding important. I have some concerns on the quality of the applied data.

We thank the reviewer for their constructive remarks and appreciation of our manuscript. We have done our best to reply to all comments addressed by the reviewer (see our replies below).

In addition, reviewer 1's remark on fire loss estimates led us to reconsider the parameter we originally used for fire severity. Consequently, we have re-calculated the fire losses component in the model parameter with coefficients of overall fire severity now ranging from 4.2 to 10.7%. This modification in the fire losses estimates implied to entirely re-run the model and re-do all figures (as well as the sensitivity analysis). This modification changed the overall result, since the estimate of global C emissions in 1990-2020 amounts to 0.74 GtC in the revised manuscript versus 2.4 GtC in the original manuscript, and down-graded the role of fire. However, it did not affect our other conclusions, in particular not the main finding on the pivotal role of changes in growth conditions and changes in forest areas.

Furthermore, reviewer 2's remark on the typology decision tree made us reconsider and improve the typology (see Fig. 3 and the 'Typology of forest biomass change' section in the main manuscript).

General comments:

Q.3.1. I cannot control every citation, but already the 2nd does not fit with the statement. Also the references used for the datasets and models do not lead to articles that used or describe the data/model. This raises serious questions on the quality of the article and need careful revision.

A.3.1. We apologize for the confusion. It was caused by the fact that we had created two reference lists: one for the main text and another one for the methods section, responding erroneously to nature communications author guidelines. It seems that the reviewer only saw the second reference list, and thus the numbers did not match. We have now merged both reference lists into one at the end of the manuscript, and thoroughly checked all references.

Q.3.2. I'm not familiar with the "FRA" dataset, but I have concerns it has the quality to be used as it was used here. It clearly needs to be validated and it needs to be proven that the dataset is reliable. I may be wrong, but for me the quality of the data may not be sufficient for a high-profile journal.

A.3.2. We thank the reviewer for this important remark regarding data reliability assessment that we have considered in our revisions. However, we would like to first answer by saying that the FRA dataset is the Forest Resource Assessment published every 5 years by the Food and Agriculture Organization of the United Nations (FAO). This is an authoritative data source which builds on national inventories and is explicitly in line with IPCC guidelines and is widely used to calibrate and validate simulations by process-based models (e.g., Bellassen et al., 2011). This database is the only one currently available containing national carbon stock data time series with global coverage. Its quality has recently been assessed by several studies and proven to be highly reliable, not to say the best available quality data (Tubiello et al., 2021; Nesha et al., 2021; FRA 2020). Please see also our answer *A.3.15* for additional details on the FRA datasets quality and uncertainties.

Bellassen et al., 2011. <https://doi.org/10.1111/j.1365-2486.2011.02476.x>

FRA 2020 <http://www.fao.org/3/ca9825en/ca9825en.pdf>

Nesha et al., 2021 <https://doi.org/10.1088/1748-9326/abd81b>

Tubiello et al., 2021 <https://doi.org/10.5194/essd-13-1681-2021>

This being said we agree with the reviewer that the reliability of the dataset needs to be discussed in the manuscript. To that end we have carried out a literature survey in order to provide independent comparisons between our modelling estimates/FRA dataset of forest biomass carbon stock changes and estimates by other studies. These comparisons have been summarized in an additional table in the Supplementary information (Table S1) and highlight that our estimations are in good agreement with the literature. In addition, we have also inserted several paragraphs in the main manuscript to appreciate the data quality and put them in context with other independent estimates.

These paragraphs read as follows:

Lines 76-86 (revised manuscript track change)

“These figures are within the range of the estimated sink in forest soil and biomass of 0.1 ± 7.3 GtC/yr in 2001-2019 found by Harris et al¹⁷. Our estimation is also consistent with that of Tubiello et al³ of 0.11 GtC/yr net C emissions from forest ecosystems. A comparison³ of FRA-derived global forest C emissions with other independent estimates reported in 1990-2015 by National Greenhouse Gas Inventories (NGHGs) – including the Russian Federation, the USA, China, Indonesia and India – and by the United Nations Framework Convention on Climate Change for other countries (UNFCCC, 2020¹⁸) yields a slight difference of c.18%, although the UNFCCC and NGHGI’s account, by definition, only for emissions from managed land³. Further independent comparisons at the national and macro-regional levels are compiled in Table S1 and reveal that C emissions estimated in the present study are in good agreements with other researches.”

References

3. Tubiello, F. N. *et al.* Carbon emissions and removals from forests: new estimates, 1990–2020. *Earth Syst. Sci. Data* **13**, 1681–1691 (2021).
17. Harris, N. L. *et al.* Global maps of twenty-first century forest carbon fluxes. *Nat. Clim. Change* (2021) doi:10.1038/s41558-020-00976-6.
18. UNFCCC: United Nations Framework Convention on Climate Change: *Time Series – Annex I, UNFCCC*. https://di.unfccc.int/time_series (Bonn).

Lines 92-104 (revised manuscript track change)

“While both harvest rate and burnt area increase globally over the period of observation, the increased forest growth rate that we calculate with CRAFT for both primary and managed forests over 1990-2020 emerges here as the only factor explaining increased biomass density at the global level. This is in line with other research pointing to the relevance of biomass thickening for forest C sequestration¹⁹. In addition, our finding that the forest growth rate increased annually by 0.19, 0.21 and 0.21 % from 1990 to 2020 respectively for primary, managed and total forests of the world is consistent with Kolby Smith et al.²⁰ who find that also Net Primary Production (NPP) increased annually between 0.10-0.25% in the period 1982-2011, as well as with other modelling and remote-sensing studies documenting a global greening trend, i.e. vegetation thickening following increased vegetation growth rate^{21,22}. Note that estimates of annual growth rate increase in 1990-2020 by the sensitivity analyses provide narrow ranges of 0.17-0.19, 0.21-0.23 and 0.20-0.22% respectively for primary, managed and total forests of the world (Table S2).”

References

19. Kauppi, P. E. *et al.* Carbon benefits from Forest Transitions promoting biomass expansions and thickening. *Glob. Change Biol.* **26**, 5365–5370 (2020).
20. Kolby Smith, W. *et al.* Large divergence of satellite and Earth system model estimates of global terrestrial CO₂ fertilization. *Nat. Clim. Change* **6**, 306–310 (2016).
21. Zhu, Z. *et al.* Greening of the Earth and its drivers. *Nat. Clim. Change* **6**, 791–795 (2016).
22. Zhao, L., Dai, A. & Dong, B. Changes in global vegetation activity and its driving factors during 1982–2013. *Agric. For. Meteorol.* **249**, 198–209 (2018).

Lines 572-578 (revised manuscript track change)

“According to the last FRA report², 86% of the forest area reported was estimated following an IPCC⁷⁶ tier 3 approach and 88% of the growing stock trend following tiers 2 or 3 approaches. Similarly, the harvest data reported by the FAO statistics are subject to several statistical tests, data validation and reconciliation with other data sources⁷⁷. The data reported by the FRA for burnt area arise from the combination of several remote-sensing approaches including the Global Wildfire Information System⁷⁸, the Moderate-Resolution Imaging Spectroradiometer (MODIS)⁷⁹ and the Global Forest Change product⁸⁰.”

References

2. *Global Forest Resources Assessment 2020*. (FAO, 2020). doi:10.4060/ca9825en.
76. Eggleston, H. S., Buendia, L., Miwa, K., Ngara, T. & Tanabe, K. *IPCC Guidelines for National Greenhouse Gas Inventories*. <http://www.ipcc-nggip.iges.or.jp/public/2006gl/index.htm> (2006).
77. Statistics Division of the FAO. *Guidelines on data collection for national statistics on forest products*. (2018).
78. Artés, T. *et al.* A global wildfire dataset for the analysis of fire regimes and fire behaviour. *Sci. Data* 6, 296 (2019).
79. Giglio, L., Boschetti, L., Roy, D. P., Humber, M. L. & Justice, C. O. The Collection 6 MODIS burned area mapping algorithm and product. *Remote Sens. Environ.* 217, 72–85 (2018).
80. Hansen, M. C. *et al.* High-Resolution Global Maps of 21st-Century Forest Cover Change. *Science* 342, 850–853 (2013).

Q.3.3. I feel that this kind of analyses done at country-level is highly biased by many factors (country size, forest area, reporting,...).

A.3.3. We thank the reviewer for this important remark regarding the scale effect on the model robustness and results reliability. To answer this remark, we made 2 types of revisions in the manuscript and SI:

- 1- We have introduced several paragraphs to better reflect the implication of our national level assessment. In addition, we further discuss the data uncertainties and we also put our results and conclusions in context with other approaches.
- 2- In order to assess the robustness of our modelling approach, we carried out 5 sensitivity analyses on the most uncertain parameters/assumptions, **including the effect of country-level data aggregation**: (i) temporal change in K parameter assumption; (ii) constant primary forest biomass density assumption; (iii) high fire impact assumption; (iv) low fire impact assumption; (v) effect of gross versus net area changes. These sensitivity analyses allow us to provide error bars where relevant in the manuscript figures (Fig. 2a & b, Fig. 3c). The main figures arising from these five sensitivity analyses are also displayed in the Supplementary Information (see Fig. S5-10). Overall the comparison between the results presented in the main manuscript *versus* those displayed in the SI as part of the sensitivity analysis confirmed the robustness of the model and the reliability of our main

conclusions: there were only differences in the magnitudes of the estimates by the various sensitivity tests but the results from the sensitivity analyses did not affect our main conclusion that changes in growth conditions more than reforestation counteracted the biomass C emissions mainly caused by deforestation in 1990-2020.

We accordingly revised the manuscript by inserting the following paragraphs in our manuscript and SI:

Lines 64-70 (revised manuscript track change, see also lines 693-731 quoted below)

Commenting on the sensitivity analysis assessment:

“The five sensitivity analyses carried out on the most uncertain model inputs and assumptions (see method descriptions and Fig.S5-10) confirmed the results presented in Fig.1: The largest deviation derived from the sensitivity analysis considering forest gross instead of net area changes results in a relative RSME of 1.94% with global C emissions c.2.5 times higher than the FRA estimates (Table S2). The simulations from the reference model assumptions yield the best RMSE and closest agreement with the C budgets derived from the FRA, indicating that they are the most optimal.”

Lines 76-86 (revised manuscript track change) *Putting our estimates in context with other studies:*

“These figures are within the range of the estimated sink in forest soil and biomass of 0.1 ± 7.3 GtC/yr in 2001-2019 found by Harris et al¹⁷. Our estimation is also consistent with that of Tubiello et al³ of 0.11 GtC/yr net C emissions from forest ecosystems. A comparison³ of FRA-derived global forest C emissions with other independent estimates reported in 1990-2015 by National Greenhouse Gas Inventories (NGHGs) – including the Russian Federation, the USA, China, Indonesia and India – and by the United Nations Framework Convention on Climate Change for other countries (UNFCCC, 2020¹⁸) yields a slight difference of c.18%, although the UNFCCC and NGHGI’s account, by definition, only for emissions from managed land³. Further independent comparisons at the national and macro-regional levels are compiled in Table S1 and reveal that C emissions estimated in the present study are in good agreements with other researches.”

References

3. Tubiello, F. N. *et al.* Carbon emissions and removals from forests: new estimates, 1990–2020. *Earth Syst. Sci. Data* **13**, 1681–1691 (2021).
17. Harris, N. L. *et al.* Global maps of twenty-first century forest carbon fluxes. *Nat. Clim. Change* (2021) doi:10.1038/s41558-020-00976-6.
18. UNFCCC: United Nations Framework Convention on Climate Change: *Time Series – Annex I, UNFCCC*. https://di.unfccc.int/time_series (Bonn).

Lines 92-104 (revised manuscript track change, *already quoted in A.2*) *Confronting our main findings with other researches:*

“While both harvest rate and burnt area increase globally over the period of observation, the increased forest growth rate that we calculate with CRAFT for both primary and managed forests over 1990-2020 emerges here as the only factor explaining increased biomass density at the global level. This is in line with other research pointing to the relevance of biomass thickening

for forest C sequestration¹⁹. In addition, our finding that the forest growth rate increased annually by 0.19, 0.21 and 0.21 % from 1990 to 2020 respectively for primary, managed and total forests of the world is consistent with Kolby Smith et al.²⁰ who find that also Net Primary Production (NPP) increased annually between 0.10-0.25% in the period 1982-2011, as well as with other modelling and remote-sensing studies documenting a global greening trend, i.e. vegetation thickening following increased vegetation growth rate^{21,22}. Note that estimates of annual growth rate increase in 1990-2020 by the sensitivity analyses provide narrow ranges of 0.17-0.19, 0.21-0.23 and 0.20-0.22% respectively for primary, managed and total forests of the world (Table S2).”

References

19. Kauppi, P. E. *et al.* Carbon benefits from Forest Transitions promoting biomass expansions and thickening. *Glob. Change Biol.* **26**, 5365–5370 (2020).
20. Kolby Smith, W. *et al.* Large divergence of satellite and Earth system model estimates of global terrestrial CO₂ fertilization. *Nat. Clim. Change* **6**, 306–310 (2016).
21. Zhu, Z. *et al.* Greening of the Earth and its drivers. *Nat. Clim. Change* **6**, 791–795 (2016).
22. Zhao, L., Dai, A. & Dong, B. Changes in global vegetation activity and its driving factors during 1982–2013. *Agric. For. Meteorol.* **249**, 198–209 (2018).

Lines 138-153 (revised manuscript track change) Commenting the sensitivity analysis assessment:

“A sensitivity analysis on the potential underestimation of C dynamics resulting from the use of net area change data at country level (see methods section and Fig. S5) reveals that accounting for gross area changes²⁶ instead of net area change would result in higher global C emissions estimates (4.19 GtC in the sensitivity test versus 0.74 GtC in the reference simulation) but would reveal the same patterns of forest C-dynamic drivers (Fig. S5). However, the magnitude of the main drivers would be slightly changed with a lower effect of changes in area (C sink in the hypothetical absence of area changes reaching 20.8 GtC in the sensitivity tests versus 26.9 GtC in the reference assessment) and a higher effect of growth rate changes (C source in the hypothetical absence of growth rate changes reaching 13.1 GtC in the sensitivity tests versus 7.4 GtC in the reference assessment). Generally, the range of results derived from the five sensitivity analyses does not change the relative importance of the individual drivers in any of the scenarios (Table S3, Fig. 2a and S5). However, the sensitivity analyses highlight that the uncertainty is large enough to reverse the cumulated C signal in the absence of changes in harvest (CF 1), changes in burnt area (CF 3), and the complete absence of burnt areas (CF 6). By contrast, the signals of CF 2 (no growth rate change), CF 4 (no area change) and CF 5 (no harvest) are larger than the uncertainty across sensitivity analyses, signaling that our findings on these drivers are most robust.”

Reference

26. Li, W. *et al.* Gross and net land cover changes in the main plant functional types derived from the annual ESA CCI land cover maps (1992–2015). *Earth Syst. Sci. Data* **10**, 219–234 (2018).

Lines 167-180 (revised manuscript track change) Reflecting on the scale issue implication:

“The fact that we use here country-level data comes both with limitations and advantages. The

main limitation associated with national data is that it conceals gross C fluxes in forest biomass dynamics and blurs heterogeneity in growth conditions and anthropogenic management within countries. The country-level resolution aggregates the effects of manifold, partly counteracting processes at the local level – including photosynthesis, maintenance respiration, growth respiration, as well as forest area loss and expansion – on the annual dynamic of primary and managed forest biomass. As a consequence, our optimization of the growth function actually reflects apparent national growth rates resulting from the aggregate of these processes. However, this simplification of forest ecosystem functioning is also an advantage. Our approach reproduces forest biomass dynamics very accurately, which is complementary to most process-based models aimed at depicting biological processes and their abiotic controls²⁷ but providing a wide range of C flux estimations¹ and hardly reproducing observation from inventory data^{1,28,29}. By contrast, the strength of the modelling approach implemented here is that it can be run with parsimonious data availability and allows to disentangle the major drivers behind forest C stock and flux trajectories.”

References

1. Friedlingstein, P. *et al.* Global Carbon Budget 2020. *Earth Syst. Sci. Data* **12**, 3269–3340 (2020).
27. Pongratz, J. *et al.* Models meet data: Challenges and opportunities in implementing land management in Earth system models. *Glob. Change Biol.* **24**, 1470–1487 (2018).
28. Bellassen, V. *et al.* Reconstruction and attribution of the carbon sink of European forests between 1950 and 2000. *Glob. Change Biol.* **17**, 3274–3292 (2011).
29. Ciais, P. *et al.* Carbon accumulation in European forests. *Nat. Geosci.* **1**, 425–429 (2008).

Lines 231-246 (revised manuscript track change) *Commenting the sensitivity analysis assessment:*

“These highlights derived from the typology remained the same in all sensitivity analyses (Fig. S6-7), despite some possible changes in country type identification (Fig. 3a and S6) and amplitude shifts in the attribution of main drivers globally (Fig. 3c and S7). The ranges of values in the attribution of main drivers result from the previously reported differences between the counterfactual and actual C budget estimates across sensitivity analyses (see also Tables S2-3) combined with some changes in the type of forest C-dynamics trajectory identified through the typology in countries with large forest biomass stocks: China, India and Australia (Fig. S6 and explanations below Fig. S6). However, these shifts do not affect the main conclusions derived from Fig. 3c: In all sensitivity analyses, growth rate changes remain the main driver of global forest biomass C sink with total net C sinks in countries where increasing growth rate is the main driver (including both solid and hatched countries) ranging from 12.1 to 21.1 GtC, while afforestation always holds the second place of global C sink driver (total net C sinks in countries where afforestation is the main driver ranging from 2.4 to 7.7 GtC). Similarly, total net C sources by countries where deforestation is the main driver range from -21.9 to -14.0 GtC, thus highlighting that deforestation would by far remain the main driver of forest biomass C emissions across all sensitivity analyses.”

Lines 693-731 (revised manuscript track change) *Describing the sensitivity analysis:*

“The robustness of the model was assessed through five sensitivity analyses carried out on the most uncertain parameters and assumptions of the model: (i) temporal change in K parameter

assumption and; **(ii)** changes in primary forest biomass density assumption; **(iii)** high fire impact assumption; **(iv)** low fire impact assumption; **(v)** effect of gross versus net area changes. As the CRAFT model reproduces the observed trends in forest biomass C stocks, its highest uncertainties lie in the estimation of the growth parameters. Here, two structural hypotheses of the model are likely to affect these calculations: **(i)** the assumption regarding the optimization of temporal changes in growth parameters and, **(ii)** the coefficient derived to estimate the temporal trend in primary forest biomass density. To assess the uncertainties associated to those assumptions, we ran 2 sensitivity analyses: **(i)** We tested an alternative hypothesis in which both the temporal trend of the r and K parameters were optimized against the reference assumption based on the optimization of the temporal change of the r parameter only **(ii)** We tested an alternative hypothesis in which biomass density in primary forest was taken as constant (using the benchmark provided in 2000 by Erb et al.¹²) against the reference assumption based on an estimation of changes in primary forest biomass density. . Burnt area is another source of uncertainty, because data are only available for the period after 2000 and for some countries missing altogether. Furthermore, it is intricate to separate burnt areas from deforestation areas. We thus consider the fire losses as the most uncertain input data to our model. Therefore, we carried out two additional sensitivity analyses: **(iii)** We tested the model with fire losses calculated by assuming the lowest fire severity coefficient (see table S3 and 4) and by assuming complete overlap between fires and deforestation in all countries (not only in tropical countries, as in the default estimation). **(iv)** We tested the model with fire losses calculated by assuming the highest fire severity coefficient (see table S4-5) assuming no overlap between forest area loss and burnt area in any countries. Last, as net area change may result in underestimating C fluxes^{81,82} we performed a sensitivity analysis **(v)** in which annual gross area gain and loss were estimated based on national conversion factor of gross to net area change derived from Li et al.²⁶ To do so, we used the data provided by Li et al.²⁶ to calculate the average net and average gross national forest area changes in 1992-2015. The ratio of net to gross area change was then derived by dividing the sum of the absolute value of all decreases and increases by the absolute value of the net change. This ratio was then used in the sensitivity analysis in order to quantify the effect of gross area change on forest biomass density (see Eq. 2). The differences in the model outputs and counterfactual scenarios assessments between the five sensitivity analyses and the default model assumptions are presented in Table S2-3 and Fig. S5-S10”

References

12. Erb, K.-H. *et al.* Unexpectedly large impact of forest management and grazing on global vegetation biomass. *Nature* 553, 73–76 (2018).
26. Li, W. *et al.* Gross and net land cover changes in the main plant functional types derived from the annual ESA CCI land cover maps (1992–2015). *Earth Syst. Sci. Data* 10, 219–234 (2018).
81. Winkler, K., Fuchs, R., Rounsevell, M. & Herold, M. Global land use changes are four times greater than previously estimated. *Nat. Commun.* **12**, 2501 (2021).
82. Fuchs, R. *et al.* Assessing the influence of historic net and gross land changes on the carbon fluxes of Europe. *Glob. Change Biol.* **22**, 2526–2539 (2016).

Lines 248-272 (revised SII track change) *Explaining the sensitivity analysis results:*

China switches from a net C sink driven by increase growth rate to a net C sink driven by reforestation in the gross area change sensitivity analysis (Fig. S6d). This is due to the fact that

both area and growth rate changes have positive effects on the net C sink in China (Fig. S3) so that changes in model assumption could result in further highlighting one driver rather than the other.

India switches from a net C source driven by afforestation to a net C source driven by increased harvest pressure in the gross area change sensitivity analysis (Fig. S6d). Considering gross area changes in India resulted in such a decrease of biomass density (due to increase rejuvenation through higher reforested area) that harvest pressure became higher than annual NPP. In such conditions, the CRAFT model could not reproduce the FRA data in Indian managed forest, which even collapsed by the end of the simulation in this sensitivity analysis, thus suggesting that the average gross-net area change ratio of c.12 in India³ might be unrealistic.

Australia switches from a net C sink driven by deforestation to a net C sink driven by increased growth rate in the dynamic K and low fire sensitivity analyses (Fig. S6b and f) but also to a net C source driven by increased fire intensity in the constant primary biomass and high fire sensitivity analyses (Fig. Sc and e). These changes in the type of C-dynamics trajectory identified in Australia are due to the fact that several drivers had large but contradicting effects in these countries: Both increased fire intensity and deforestation contributed to a net C source while increased growth rate was actually the largest driver counteracting the C source in the reference model simulation (Fig. S3). In those conditions, it is unsurprising that changes in the model assumption and parameters are more likely to affect the C budget and main driver attribution in this country.

In addition, it is worth noting that many West European countries switch from a net C sink driven by increased growth rate to a net C sink driven by reforestation and *vice-versa* across the different sensitivity analyses. These changes result from the synergetic effects of both reforestation and growth rate changes in these countries (Fig. S3), so that changes in model assumption may result in further highlighting one driver rather than the other.

Title:

Q.3.4. biomass carbon emissions from what? disturbances?

A.3.4. We modified the title as follows to make it clearer: “Altered growth conditions more than reforestation counteracted forest biomass carbon emissions 1990-2020”

Abstract:

Q.3.5. It is not totally clear what has been done and what the main message is.

I’m not familiar with a “counterfactual modelling approach”. Was observational data used, so are the results observed reality or rather theoretical? That also applies to the title, not clear at this point if this is something that happened (like suggested in the title), or simulated

A.3.5. The abstract is very limited in words. Therefore, we do not have enough space to develop here on the counterfactual modelling approach. We provide more explanation later in the text and in the method section to explain what it is and how we made it (see citations below)

Regarding the dataset and modelling approach, we used the Forest Resource Assessment (FRA)

datasets and integrate them in the CRAFT model. The FRA contains information on biomass C-stocks, harvest and forest area changes at national level and annual resolution, but this information is not sufficient to quantify the relative importance of specific drivers, because forest growth information is required to close the balance between the dynamics of these three parameters. CRAFT builds upon the establishment of a place and time specific relationship between net primary production (NPP) and biomass C stocks and so provides a suitable tool to systematically integrate the available information. In a nutshell, CRAFT reliably reproduces the observed trends in forest biomass C stocks while adding information on annual estimations of forest C stocks, rather than the 5-years interval data provided by the FRA, and disentangling the relative impact of various drivers on forest change.

Regarding the “counterfactual modelling approach”, it is a method based on comparisons between actual and counterfactual trajectories aimed at quantifying the effect of an isolated factor on the trajectory investigated. In the present study the actual trajectories of forest biomass C stocks are assessed through both FRA datasets and the CRAFT model (which allows to reproduce the trends reported by the FRA) while the counterfactual trajectories are estimated by keeping one of the model parameters fixed at its initial value (e.g., constant forest area). Then, the actual biomass C budgets in the 1990-2020 period is compared with those simulated in different counterfactual scenarios (see e.g., Fig. 2a).

Please also refer to the following paragraph in the main manuscript:

Lines 483-488 (revised manuscript track change) *Regarding the modelling approach:*

“CaRbon Accumulation in ForesT (CRAFT) is a parsimonious model of long-term changes in carbon (C) stocks and fluxes in forest ecosystems¹⁶. CRAFT builds upon the establishment of a place and time specific relationship between NPP and biomass C stocks and so provides a suitable tool to simulate C stock dynamics in forest biomass and soils (including both above-ground and belowground biomass).”

Reference

16. Le Noë, J. *et al.* Modeling and empirical validation of long-term carbon sequestration in forests (France, 1850–2015). *Glob. Change Biol.* 26, 2421–2434 (2020).

Lines 114-120 (revised manuscript track change) *Regarding the counterfactual approach:*

“We develop six counterfactual scenarios^{23–25} in order to investigate how forest biomass density and forest biomass C stocks would evolve in the hypothetical absence of: (i) changes in harvest (**CF1**); (ii), changes in forest growth rates (**CF2**); (iii) change in burnt area (**CF 3**); (iv) change in forest area (**CF 4**); (v) harvest (**CF5**); (vi) burnt area (**CF 6**) (see method section). The comparison of observed and simulated counterfactual trends allows us to isolate and quantify the influence of these four main drivers on global forest C stock changes at national resolution (CF1 to 4) as well as to quantify the overall effects of total wood extraction and burnt area (CF5 and 6).”

References

23. Hong, C. *et al.* Global and regional drivers of land-use emissions in 1961–2017. *Nature*

589, 554–561 (2021).

24. Burney, J. A., Davis, S. J. & Lobell, D. B. Greenhouse gas mitigation by agricultural intensification. *Proc. Natl. Acad. Sci.* 107, 12052–12057 (2010).

25. Gingrich, S., Lauk, C., Krausmann, F., Erb, K.-H. & Le Noë, J. Changes in energy and livestock systems largely explain the forest transition in Austria (1830–1910). *Land Use Policy* 109, 105624 (2021).

Lines 196-207 (revised manuscript track change) *Regarding the typology based on counterfactual scenario approach:*

“In order to identify spatial and temporal patterns of drivers in forest biomass trends, we establish a typology of the main drivers over the period 1990-2020 (Fig. 3b). The typology we established is based on the positive versus negative shift in biomass C stocks, and highlights the most important driver of this shift as assessed through the counterfactual assessment, irrespective of the relative importance of the other drivers shown in Fig. 2. However, as the early separation between increasing and decreasing biomass C stocks in the decision tree (Fig. 3b) may conceal the effect of a major driver counteracting the observed C dynamic, the typology also accounts for possible antagonistic effects by identifying cases in which the main driver of observed C dynamics is not, in absolute terms, the most important driver (e.g., C stocks increase but the driver with the strongest absolute effect counteracts this positive budget, see also Fig. S3). By pinpointing the major drivers of forest change at national levels, such an approach enables to identify major levers for forest conservation.”

Lines 646-657 (revised manuscript track change) *Regarding the counterfactual approach (method section):*

“In order to isolate and quantify the relative impacts of major proximate drivers on forest biomass C change, we develop six counterfactual scenarios^{23-25,75}. In these counterfactual scenarios we investigate how forest biomass density and forest biomass C stocks would evolve in the hypothetical absence of: (i) change in harvest (assuming initial average harvest volumes to remain constant; **CF1**); (ii), change in forest growth rate (calculated values for 1990 remain constant; **CF2**); (iii) change in burnt area (average values for 2000 to 2003 are assumed to remain constant ; note that FRA does not report burnt area before 2000; **CF 3**); (iv) change in forest area (values for 1990 remain constant; **CF 4**); (v) harvest (no-harvest counterfactual; **CF5**); (vi) fire occurrence (no burnt area counterfactual; **CF 6**). The comparison of observed and simulated trends allows to isolate and quantify the influence of the four main drivers on the global C stock changes at national resolution (CF1 to 4) as well as to quantify the overall effects of total wood extraction and fire occurrence (CF5 and 6).”

References

23. Hong, C. *et al.* Global and regional drivers of land-use emissions in 1961–2017. *Nature* 589, 554–561 (2021).

24. Burney, J. A., Davis, S. J. & Lobell, D. B. Greenhouse gas mitigation by agricultural intensification. *Proc. Natl. Acad. Sci.* 107, 12052–12057 (2010).

25. Gingrich, S., Lauk, C., Krausmann, F., Erb, K.-H. & Le Noë, J. Changes in energy and livestock systems largely explain the forest transition in Austria (1830–1910). *Land Use Policy* 109, 105624 (2021).

75. Meyfroidt, P. Approaches and terminology for causal analysis in land systems science. *J.*

Land Use Sci. **11**, 501–522 (2016).

Q.3.6. line 14: C emissions from fossil fuel burning? or harvest, fire?

A.3.6. These are C emissions from forest biomass C stock changes. We have now modified the abstract to make it clearer:

Lines 7-21 (revised manuscript track change)

“Understanding the carbon (C) balance in global forest is key for climate-change mitigation. However, land use and environmental drivers affecting global forest C fluxes remain poorly quantified. Following a counterfactual modelling approach based on global Forest Resource Assessments, we show that in 1990-2020 deforestation was the main driver of forest C emissions, partly counteracted by increased forest growth rates under altered conditions: In the hypothetical absence of changes in forest (i) area, (ii) harvest or (iii) burnt area, global forest biomass would have reversed from an actual cumulative net C source of c. 0.74 GtC to a net C sink of 26.9, 4.9 and 0.63 GtC, respectively. In contrast, (iv) without growth rate changes, cumulative emissions would be 7.4 GtC, i.e. 10 times higher. Because this sink function may be discontinued in the future due to climate-change, ending deforestation and lowering wood harvest emerge as key climate-change mitigation strategies.”

Q.3.7. there are no numbers supporting the reforestation statement

A.3.7. We are very limited in abstract word counts so we focus on the most relevant figures in the abstract:

Lines 12-15 (revised manuscript track change)

“In the hypothetical absence of changes in forest (i) area, (ii) harvest or (iii) burnt area, global forest biomass would have reversed from an actual cumulative net C source of c. 0.74 GtC to a net C sink of 26.9, 4.9 and 0.63 GtC, respectively. In contrast, (iv) without growth rate changes, cumulative emissions would be 7.4 GtC, i.e. 10 times higher.”

In addition, we also provide numbers later in the text to support the reforestation statement in the abstract:

Lines 122-126 (revised manuscript track change):

“In the absence of changes in area, global forest biomass would act as a cumulative net C sink of c. 26.9 GtC in the study period, creating a difference of 27.6 GtC between the actual and the CF4 C budget. This effect in the absence of area change, however, is a composite of an additional C sink of 30.7 in deforesting countries and an additional C source of 3.8 GtC in reforesting countries.”

Q.3.8. why may the sink be discontinued in the future?”

A.3.8. Again, as the word count is very limited in the abstract, we further elaborate and improve this in the discussion of the paper saying the sink promoted by growth rate acceleration may be discontinued in the future because:

Lines 313-320 (revised manuscript track change):

“Research suggests that increasing forest growth rate is a transient phenomenon and might be discontinued in the future⁴⁶. For instance, several recent studies have pointed towards the saturating effect of CO₂ fertilization, which is suspected to be a key process underlying vegetation greening and ensuing thickening²¹, the risk of increasing mortality and slower growth rate following increasing drought^{6,47,48}, temperature⁴⁹ and natural disturbances such as insect outbreaks^{50,51}. Even more recently, Duffy et al.⁵² showed that, in the near-future, temperature increases from business-as-usual trajectories of climate change shall result in a severe reduction, and possibly a reversal, of the terrestrial C sink, despite the remaining unknowns.”

References

6. Hubau, W. *et al.* Asynchronous carbon sink saturation in African and Amazonian tropical forests. *Nature* 579, 80–87 (2020).
21. Zhu, Z. *et al.* Greening of the Earth and its drivers. *Nat. Clim. Change* 6, 791–795 (2016).
46. Pan, N. *et al.* Increasing global vegetation browning hidden in overall vegetation greening: Insights from time-varying trends. *Remote Sens. Environ.* 214, 59–72 (2018).
47. McDowell, N. G. *et al.* Pervasive shifts in forest dynamics in a changing world. *Science* 368, eaaz9463 (2020).
48. Senf, C., Buras, A., Zang, C. S., Rammig, A. & Seidl, R. Excess forest mortality is consistently linked to drought across Europe. *Nat. Commun.* 11, 6200 (2020).
49. Senf, C. *et al.* Canopy mortality has doubled in Europe’s temperate forests over the last three decades. *Nat. Commun.* 9, 4978 (2018).
50. Seidl, R. *et al.* Forest disturbances under climate change. *Nat. Clim. Change* 7, 395–402 (2017).
51. Nabuurs, G.-J. *et al.* First signs of carbon sink saturation in European forest biomass. *Nat. Clim. Change* 3, 792–796 (2013).
52. Duffy, K. A. *et al.* How close are we to the temperature tipping point of the terrestrial biosphere? *Sci. Adv.* 7, eaay1052 (2021).

Introduction:

Q.3.9. More than half of the introduction is method description, that’s quite an imbalance.

A.3.9. We have shortened the brief method description as far as possible which now stands in only three sentences (see revisions reported just below). This mix is also a constraint arising from the Nature writing style according to which the method description should stand at the end of the manuscript and as we still wish the general approach to be clear for readers who won’t read the extensive method section we very briefly state our methods.

Lines 36-49 (revised manuscript track change):

“Here, we fill that gap by combining the most recent and consistent global forest dataset provided by the Forest Resource Assessment (FRA²) – an authoritative data source^{3,15} – with the parsimonious forest C model CRAFT¹⁶ (CaRbon Accumulation in ForesT). This enables us to isolate and quantify the relative impact of various drivers on forest change, including, for the first time, changes in forest growth rates resulting from altered growth conditions. We calculate

the temporal dynamics of managed and primary (i.e. unmanaged) forest growth rates in 152 countries in 1990-2020 and couple counterfactual scenario development with a typology approach (see method section) in order to answer the following question”

References

2. *Global Forest Resources Assessment 2020*. (FAO, 2020). doi:10.4060/ca9825en.
3. Tubiello, F. N. *et al.* Carbon emissions and removals from forests: new estimates, 1990–2020. *Earth Syst. Sci. Data* 13, 1681–1691 (2021).
15. Nesha, M. K. *et al.* An assessment of data sources, data quality and changes in national forest monitoring capacities in the Global Forest Resources Assessment 2005-2020. *Environ. Res. Lett.* (2021) doi:10.1088/1748-9326/abd81b.
16. Le Noë, J. *et al.* Modeling and empirical validation of long-term carbon sequestration in forests (France, 1850–2015). *Glob. Change Biol.* 26, 2421–2434 (2020).

Q.3.10. 1. The introduction starts with a strong statement: “global forest biomass has acted as a net C source to the atmosphere over the last three decades, according to the most recent forest resource assessment”. It is referred with an old paper from 2002 “Increasing carbon stocks in the forest soils of western Europe”, which does fit with the “three decades and most recent forest assessment”, and is also not really relate with this statement. I do not want to “control” each sentence, statement and how it is referenced, this is clearly the responsibility of the authors, but this is a high-level journal and this needs to be done more careful...

I randomly checked another reference, number 10 does not really relate with “forest biomass C dynamics” as stated in the sentence. I’m normally not picky with this, but at this journal level, references need to be precise.

A.3.10. Again, we apologize we created two reference lists which created this. We have now merged both lists and the referencing are now correct. Please see also our answer A.3.1. above.

Q.3.11. 2. Again, the most important thing, the data source, is not cited correctly. I do not find a “global Forest Resource Assessment” under ref 2 or 13.

A.3.11. Please see also our answer A.3.1. above.

Q.3.12. 3. Again, the model used is referenced with the article “Classifying drivers of global forest loss” (ref 14), which does however neither use or describe the model.

A.3.12. Please see also our answer A.3.1. above.

Q.3.14. 4. line 35: what is FRA? this was never spelled out or explained.

A.3.14. We apologize for never spelling before, this is now corrected (FRA = Forest Resource Assessment by the Forest and Agriculture Organization).

Results:

Q.3.15. 1. line 54: “The CRAFT model reliably reproduces the observed trends in primary and production forest biomass C stocks (including both aboveground and belowground biomass) in 1990-2020 with a relative root mean square error (RMSE) between simulated and observed biomass C stocks.” We need to know what data feeds into the model and what are used for validating. How can you observe biomass C stocks 1990-2020? Is this the FRA dataset? I never heard about it before, what is it based on? a biomass time series 1990-2020 sounds too good to be true. How can it observe belowground biomass at global scale? I have doubts on the FRA dataset to be used for such a strong conclusion as the paper does, the method section does not really give sufficient information on its quality. The data seems to be country based? Isnt this highly biased by the country reporting, the forest area, etc? The section on data uncertainty does not help much to get trust in the data.

A.3.15. We thank the reviewer for this remark regarding dataset reliability and model robustness. We have largely answered those remarks in our previous answers (see A.3.2 and A.3.3).

To further clarify, the FRA database is an authoritative source of input data on forest surface, harvest, standing biomass and burnt area. As a matter of fact, this database arises from the Food and Agriculture Organization of the United Nations (FAO) which collects, analyses and disseminates at least every 5-years since 1990 a country-based forest statistic, describing the status of forests with data at the country, regional and global level. This database, providing information on forest land area and carbon stock has been used to provide critical inputs to the main scientific consortium on Global Change and Climate change, including the work by Houghton and Nassikas (2017) as well as the Global C budget project (Friedlingstein et al., 2020) which are used for the IPCC report. Furthermore, analyses of the FRA dataset and comparisons with alternatives for the year 2000 (see Erb et al. 2018), revealed its reliability, in particular for temperate and boreal regions, but also for tropical regions, also if for the latter uncertainties are larger in the absence of data (Erb et al., 2018). While these analyses relate to the year 2000, the uncertainty related to C-stock trends cannot be assessed due to a lack of alternative data; however, FRA is used for many studies as benchmark dataset for carbon stock (e.g. Baccini et al., 2012, Spawn et al., 2020, Tubiello et al., 2021, Xu et al., 2021) and forest area changes (e.g. Song et al., 2018, Li et al., 2018). As a side-note, the dataset by Spawn et al., (2020) also relies on FRA data and approaches for assessing below-ground biomass. Therefore, we argue that the data quality, although associated to unavoidable uncertainties, is nonetheless the best and most recent data which could be used. Indeed, Tubiello et al., (2021) recently published a paper in which the uncertainties associated to the FRA assessment were appreciated by comparing these net C emissions from national and global forest by the FRA with National GHG inventory report and with the UNFCCC estimations. This comparison revealed that these independent estimations were in very close agreements, thus confirming the reliability of the FRA database.

Baccini et al., (2012). <https://doi.org/10.1038/nclimate1354>

Erb et al. (2018). <https://doi.org/10.1038/nature25138>

Friedlingstein et al., (2020) <https://doi.org/10.5194/essd-12-3269-2020>

Houghton and Nassikas (2017). <https://doi.org/10.1002/2016GB005546>

Li et al., (2021). <https://doi.org/10.5194/essd-10-219-2018>

Spawn et al., (2020). <https://doi.org/10.1038/s41597-020-0444-4>

Song et al., (2018) <https://doi.org/10.1038/s41586-018-0411-9>
Tubiello et al., (2021). <https://doi.org/10.5194/essd-13-1681-2021>
Xu et al., (2021). <https://doi.org/10.1126/sciadv.abe9829>

Discussion on dataset reliability have been inserted in the revised manuscript. *Please, see Lines 76-86, 92-104 and 572-578 (revised manuscript track change) quoted in our previous answer A.3.2 on uncertainty and data quality discussion.* However, the situation is different with regard to fire. Even though we argue that the FRA and FAO stat provides the best time-series available on National forest, we acknowledge that losses by fire are currently the most uncertain input data derived from this dataset. As a consequence, we performed two sensitivity analyses assuming low and high fire loss estimates (in addition to three other sensitivity analyses). We have added the following paragraph in the main manuscript and added Fig in the SI to support our findings. *Please, see the lines 693-731 (revised manuscript track change) regarding the sensitivity analyses performed on fire losses estimates quoted in our previous answer A.3.3.*

Q.3.16. 2. Similar concerns apply on data on harvest and fire. For me, independent validation data are needed to support the conclusions.

A.3.16. Regarding harvest data, the FAO stat report provides a several comparison and data reconciliation with other data sources, which makes it the most important dataset centralizing inventory and statistic from all over the world. We added the following sentence:

Lines 674-676 (revised manuscript track change, already quoted in A.3.2):

“Similarly, the harvest data reported by the FAO statistics are subject to several statistical tests, data validation and reconciliation with other data sources⁷⁷.”

References

77. Statistics Division of the FAO. *Guidelines on data collection for national statistics on forest products.* (2018).

Estimation of burnt area by the FAO are already the result of comparison and reconciliation of various remote sensing approach. We added the following sentence:

Lines 676-678 (revised manuscript track change, already quoted in A.3.2):

“The data reported by the FRA for burnt area arise from the combination of several remote-sensing approaches including the Global Wildfire Information System⁷⁸, the Moderate-Resolution Imaging Spectroradiometer (MODIS)⁷⁹ and the Global Forest Change product⁸⁰.”

References

78. Artés, T. *et al.* A global wildfire dataset for the analysis of fire regimes and fire behaviour. *Sci. Data* 6, 296 (2019).

79. Giglio, L., Boschetti, L., Roy, D. P., Humber, M. L. & Justice, C. O. The Collection 6 MODIS burned area mapping algorithm and product. *Remote Sens. Environ.* 217, 72–85 (2018).

80. Hansen, M. C. *et al.* High-Resolution Global Maps of 21st-Century Forest Cover Change. *Science* 342, 850–853 (2013).

Regarding the remark on data validation we would like to highlight that:

- Our model actually reproduces the FRA datasets which are the best available quality data (see our comments in A.3.2 and A.3.15)
- Both model estimates and FRA datasets are now compared with other independent researches on forest biomass C budgets (see lines 76-86 in the revised manuscript track change, Table S1 in the revised SI1 track change, *also quoted in A.3.2*).
- We extended the discussion on the uncertainty related to the FRA and FAO data from which we derived most of the input data for our modelling approach (see lines 572-578 in the revised manuscript track change, *also quoted in A.3.2*).
- We now confronted our main findings – changes in growth conditions are key driver counteracting forest C emissions – with other modelling studies. To that end we carried additional literature survey and inserted several sentences or paragraphs emphasizing that other modelling and remote-sensing studies also found that changes in growth conditions resulted in longer growing season and vegetation greening (see lines 95-104 in the revised manuscript track change, *also quoted in A.3.3*). However, we also point that contrary to existing study, our simple modelling approach can be run with parsimonious data availability and allows to disentangle the major drivers behind forest C stock and flux trajectories.
- We carried out additional sensitivity analyses (including the effect of gross/net area change, high and low estimates of fire losses) which allowed us to better evaluate the robustness of our modelling approach and to provide error bar where relevant (see lines 64-70, 138-153, 231-246, 693-731 in the revised manuscript track change, Tables S2-3, Fig. S5-10 and lines 248-272 in the revised SI track change, *also quoted in A.3.3*)

REVIEWERS' COMMENTS

Reviewer #1 (Remarks to the Author):

The authors have done a fantastic and attentive job of addressing my concerns. I approve of publication of this manuscript in Nature Communications. Thank you for the very explicit and thorough response!

Reviewer #2 (Remarks to the Author):

I would like to thank the authors for the thorough revision of their manuscript. They have addressed my comments, and I consider the manuscript to be of great interest to the research community. The use of national statistics in here advances new analyses in the field of research, despite the disadvantages and uncertainties that arise from these types of data.

I am glad with the revision of the typology; I understand that the authors want to maintain net increase/decrease information from the colour scale, and think that the use of hatching is a neat way to convey that the largest driver may have an opposite signal.

I have picked up a few mistakes in the revised manuscript that I list below:

L. 50: "results" should read "result"

L. 58: "RSME" should read "RMSE"

L. 77: "researches": consider replacing with "research" or "studies"

Supp. Mat. L. 326: remove "in"

Table S9: consider adding "of": "Summary of all the available..."

REVIEWERS' COMMENTS

Reviewer #1 (Remarks to the Author):

The authors have done a fantastic and attentive job of addressing my concerns. I approve of publication of this manuscript in Nature Communications. Thank you for the very explicit and thorough response!

We are very grateful to the reviewer for his/her nice reply.

Reviewer #2 (Remarks to the Author):

I would like to thank the authors for the thorough revision of their manuscript. They have addressed my comments, and I consider the manuscript to be of great interest to the research community. The use of national statistics in here advances new analyses in the field of research, despite the disadvantages and uncertainties that arise from these types of data.

I am glad with the revision of the typology; I understand that the authors want to maintain net increase/decrease information from the colour scale, and think that the use of hatching is a neat way to convey that the largest driver may have an opposite signal.

We are very grateful to the reviewer for his/her nice reply.

I have picked up a few mistakes in the revised manuscript that I list below:

L. 50: "results" should read "result"

L. 58: "RSME" should read "RMSE"

L. 77: "researches": consider replacing with "research" or "studies"

Supp. Mat. L. 326: remove "in"

Table S9: consider adding "of": "Summary of all the available..."

We thank the reviewer for his/her careful reading. We have corrected all the mistakes pointed out in the comments